# IF YOUR DATA DISTRIBUTION SHIFTS, USE SELF-LEARNING

## ABSTRACT

We demonstrate that self-learning techniques like entropy minimization and pseudo-labeling are simple and effective at improving performance of a deployed computer vision model under systematic domain shifts. We show consistent improvements irrespective of the model architecture, the pre-training technique or the type of distribution shift. At the same time, self-learning is simple to use in practice because it does not require knowledge or access to the original training data or scheme, is robust to hyperparameter choices, is straight-forward to implement and requires only a few adaptation epochs. This makes self-learning techniques highly attractive for any practitioner who applies machine learning algorithms in the real world. We present state-of-the art adaptation results on CIFAR10-C (8.5% error), ImageNet-C (22.0% mCE), ImageNet-R (17.4% error) and ImageNet-A (14.8% error), theoretically study the dynamics of self-supervised adaptation methods and propose a new classification dataset (ImageNet-D) which is challenging even with adaptation.

## 1 INTRODUCTION

Deep Neural Networks (DNNs) can reach human-level performance in complex cognitive tasks (Brown et al., 2020; He et al., 2016a; Berner et al., 2019) if the distribution of the test data is sufficiently similar to the training data. However, DNNs are known to struggle if the distribution of the test data is shifted relatively to the training data (Geirhos et al., 2018; Dodge & Karam, 2017).

Two largely distinct communities aim to increase the performance of models under test-time distribution shifts: The *robustness community* generally considers ImageNet-scale datasets and evaluates models in an *ad-hoc* scenario. Models are trained on a clean source dataset like ImageNet, using heavy data augmentation (Hendrycks et al., 2020a; Rusak et al., 2020; Geirhos et al., 2019) and/or large-scale pre-training (Xie et al., 2020a; Mahajan et al., 2018). The trained models are not adapted in any way to test-time distribution shifts. This evaluation scenario is relevant for applications in which very different distribution shifts are encountered in an unpredictable order, and hence misses out on the gains of adaptation to unlabeled samples of the target distribution.

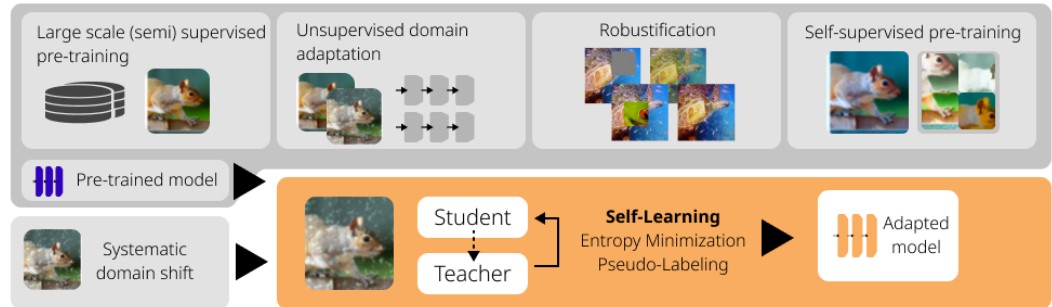

Figure 1: Robustness and adaptation to new datasets has traditionally been achieved by robust pre-training (with hand-selected/data-driven augmentation strategies, or additional data), unsupervised domain adaptation (with access to unlabeled samples from the test set), or, more recently, self-supervised learning methods. We show that on top of these different pre-training tasks, it is always possible (irrespective of architecture, model size or pre-training algorithm) to further adapt models to the target domain with simple self-learning techniques.

The *unsupervised domain adaptation (UDA) community* often considers smaller-scale datasets and assumes that both the source and the (unlabeled) target dataset are known. Models are trained on both datasets (e.g., with an adversarial domain objective, Ganin et al., 2016) before evaluation on the target domain data. This evaluation scenario provides optimal conditions for adaptation, but the reliance on the source dataset makes UDA more computationally expensive, more impractical and prevents the use of pre-trained models for which the source dataset is unknown or simply too large.

In this work, we consider the *source-free domain adaptation setting*, a middle ground between the classical ad-hoc robustness setting and UDA in which models can adapt to the target distribution but without using the source dataset (Kundu et al., 2020; Kim et al., 2021; Li et al., 2020; Liang et al., 2020). This evaluation scenario is interesting for many practitioners and applications as an extension of the ad-hoc robustness scenario. It evaluates the possible performance of a *deployed* model on a systematic, unseen distribution shift at inference time: an embedded computer vision system in an autonomous car should adapt to changes without being trained on all available training data; an image-based quality control software may not necessarily open-source the images it has been trained on, but still has to be adapted to the lighting conditions at the operation location; a computer vision system in a hospital should perform robustly when tested on a scanner different from the training images—importantly, it might not be known at development time which scanner it will be tested on, and it might be prohibited to share images from many hospitals to run UDA.

Can self-learning methods like *pseudo-labeling* and *entropy-minimization* also be used in this *source-free* domain adaptation setting? To answer this question, we perform an extensive study of several self-learning variants, and find consistent and substantial gains in test-time performance across several robustness and out-of-domain benchmarks and a wide range of models and pre-training methods, including models trained with UDA methods that do not use self-learning. We also find that self-learning outperforms state-of-the-art source-free domain adaptation methods, namely Test-Time Training which is based on a self-supervised auxiliary objective and continual training (Sun et al., 2019b), test-time entropy minimization (Wang et al., 2020) and (gradient-free) BatchNorm adaptation (Schneider et al., 2020; Nado et al., 2020). We perform a large number of ablations to study important design choices for self-learning methods in source-free domain adaptation. Furthermore, we show that a variant of pseudo-labeling with a robust loss function consistently outperforms entropy minimization on ImageNet-scale datasets. We theoretically analyze and empirically verify the influence of the temperature parameter in self-learning and provide guidelines how this single parameter should be chosen. Our approach is visualized in Figure 1. We do not consider test-time adaptation in an online setting like is studied e.g., by Zhang et al. (2021), where the model is adapted to one example at a time, and reset after each example.

**Related Work.** Variants of self-learning have been used for UDA (Berthelot et al., 2021), for example using auxiliary information (Xie et al., 2020b), consistency (Wei et al., 2020; Cai et al., 2021; Prabhu et al., 2021) or confidence (Zou et al., 2019) regularization. The main difference from these works to ours is that they 1) utilize both source and target data for self-learning whereas we only require access to unlabeled target data, 2) train their models from scratch whereas we merely fine-tune pretrained checkpoints on the unlabeled target data, and 3) are generally more complicated than our approach due to using more than one term in the objective function.

Our work is conceptually most similar to virtual adversarial domain adaptation in the fine-tuning phase of DIRT-T (Shu et al., 2018)) and Test-time entropy minimization (TENT; Wang et al., 2020). In contrast to DIRT-T, our objective is simpler and we scale the approach to considerably larger datasets on ImageNet scale. TENT, on the other hand, only evaluated a single method (entropy minimization) on a single vanilla model (ResNet-50) on IN-C. We substantially expand this analysis to show that self-learning almost universally increases test-time performance under distribution shifts, regardless of the type of distribution shift, the model architecture or the pre-training method.

Self-learning has also been applied to UDA for semantic segmentation (Zou et al., 2018), for gradual domain adaptation (Kumar et al., 2020), for semi-supervised learning (Rizve et al., 2021; Mukherjee & Awadallah, 2020), for learning in biased datasets (Chen et al., 2020b) and for automated data annotation (De Sousa Ribeiro et al., 2020). Zoph et al. (2020) show that self-learning outperforms pretraining when stronger data augmentation is used and more labeled data is present. A more detailed discussion of related work alongside with the main differences to our work can be found in Appendix F. Our main contribution beyond these works is to show the effectiveness of self-learning on top of both robust, large scale, and domain adapted models, at scale.

## 2 SELF-LEARNING FOR TEST-TIME ADAPTATION

Different variants of self-learning have been used in both unsupervised domain adaptation (French et al., 2018; Shu et al., 2018), self-supervised representation learning (Caron et al., 2021), and in semi-supervised learning (Xie et al., 2020a). In a typical self-learning setting a *teacher* network $\mathbf{f}^t$ trained on the source domain predicts labels on the target domain. Then, a *student* model $\mathbf{f}^s$ is fine-tuned on the predicted labels.

In the following, let $\mathbf{f}^t(\mathbf{x})$ denote the logits for sample $\mathbf{x}$ and let $p^t(j|\mathbf{x}) \equiv \sigma_j(\mathbf{f}^t(\mathbf{x}))$ denote the probability for class $j$ obtained from a softmax function $\sigma_j(\cdot)$. Similarly, $\mathbf{f}^s(\mathbf{x})$ and $p^s(j|\mathbf{x})$ denote the logits and probabilities for the student model $\mathbf{f}^s$. For all techniques, one can optionally only admit samples where the probability $\max_j p^t(j|\mathbf{x})$ exceeds some threshold. We consider three popular variants of self-learning: Pseudo-labeling with hard or soft labels, as well as entropy minimization.

**Hard Pseudo-Labeling (Lee, 2013; Galstyan & Cohen, 2007).** We generate labels using the teacher and train the student on pseudo-labels $i$ using the standard cross-entropy loss,

$$\ell_H(\mathbf{x}) := -\log p^s(i|\mathbf{x}), \quad i = \mathrm{argmax}_j \, p^t(j|\mathbf{x}) \tag{1}$$

Usually, only samples with a confidence above a certain threshold are considered for training the student. We test several thresholds but note that thresholding means discarding a potentially large portion of the data which leads to a performance decrease in itself. The teacher is updated after each epoch.

**Soft Pseudo-Labeling (Lee, 2013; Galstyan & Cohen, 2007).** In contrast to the hard pseudo-labeling variant, we here train the student on class probabilities predicted by the teacher,

$$\ell_S(\mathbf{x}) := -\sum_j p^t(j|\mathbf{x}) \log p^s(j|\mathbf{x}). \tag{2}$$

Soft pseudo-labeling is typically not used in conjunction with thresholding, since it already incorporates the certainty of the model. The teacher is updated after each epoch.

**Entropy Minimization (ENT; Grandvalet & Bengio, 2004).** This variant is similar to soft pseudo-labeling, but we no longer differentiate between a teacher and student network. It corresponds to an "instantaneous" update of the teacher. The training objective becomes

$$\ell_E(\mathbf{x}) := -\sum_j p^s(j|\mathbf{x}) \log p^s(j|\mathbf{x}). \tag{3}$$

Intuitively, self-training with entropy minimization leads to a sharpening of the output distribution for each sample, making the model more confident in its predictions.

**Robust Pseudo-Labeling (RPL).** Virtually all introduced self-training variants use the standard cross-entropy classification objective. However, the standard cross-entropy loss has been shown to be sensitive to label noise (Zhang & Sabuncu, 2018; Zhang et al., 2017). In the setting of domain adaptation, inaccuracies in the teacher predictions and, thus, the labels for the student, are inescapable, with severe repercussions for training stability and hyperparameter sensitivity as we show in the results.

As a straight-forward solution to this problem, we propose to replace the cross-entropy loss by a robust classification loss designed to withstand certain amounts of label noise (Ghosh et al., 2017; Song et al., 2020; Shu et al., 2020; Zhang & Sabuncu, 2018). A popular candidate is the *Generalized Cross Entropy (GCE)* loss which combines the noise-tolerant Mean Absolute Error (MAE) loss (Ghosh et al., 2017) with the CE loss. We only consider the hard labels and use the robust GCE loss as the training loss for the student,

$$i = \mathrm{argmax}_j \, p^t(j|\mathbf{x}), \quad \ell_{GCE}(\mathbf{x}, i) := q^{-1}(1 - p^s(i|\mathbf{x})^q), \tag{4}$$

with $q \in (0, 1]$. For the limit case $q \to 0$, the GCE loss approaches the CE loss and for $q = 1$, the GCE loss is the MAE loss (Zhang & Sabuncu, 2018). We test updating the teacher both after every update step of the student (RPL) and once per epoch (RPL^ep).

## 3 EXPERIMENT DESIGN

**Datasets.** IN-C (Hendrycks & Dietterich, 2019) contains corrupted versions of the $50\,000$ images in the IN validation set. There are fifteen test and four hold-out corruptions, and there are five severity levels for each corruption. The established metric to report model performance on IN-C is the mean Corruption Error (mCE) where the error is normalized by the AlexNet error, and averaged over all corruptions and severity levels, see Eq. 20, Appendix C.1. IN-R (Hendrycks et al., 2020a) contains $30\,000$ images with artistic renditions of 200 classes of the IN dataset. IN-A (an, 2019) is composed of 7500 unmodified real-world images on which standard IN-trained ResNet50 (He et al., 2016b) models yield chance level performance. CIFAR10 (Krizhevsky et al., 2009) and STL10 (Coates et al., 2011) are small-scale image recognition datasets with 10 classes each, and training sets of $50\,000$/5000 images and test sets of $10\,000$/8000 images, respectively. The digit datasets MNIST (Deng, 2012) and MNIST-M (Ganin et al., 2016) both have $60\,000$ training and $10\,000$ test images.

**Hyperparameters.** The different self-learning variants have a range of hyperparameters such as the learning rate or the stopping criterion. Our goal is to give a realistic estimation on the performance to be expected in practice.. To this end, we optimize hyperparameters for each variant of pseudo-labeling on a hold-out set of IN-C that contains four types of image corruptions ("speckle noise", "Gaussian blur", "saturate" and "spatter") with five different strengths each, following the procedure suggested in Hendrycks & Dietterich (2019). We refer to the hold-out set of IN-C as our *dev* set.

**Models for ImageNet-scale datasets.** We consider four popular model architectures: ResNet50 (He et al., 2016b), DenseNet161 (Huang et al., 2017), ResNeXt101 (Xie et al., 2017) and EfficientNet-L2 (Tan & Le, 2019) (see Appendix B.1 for details on the used models). For ResNet50, DenseNet and ResNeXt101, we include a simple *vanilla* version trained on IN only. For ResNet50 and ResNeXt101, we additionally include a state-of-the-art robust version trained with DeepAugment and Augmix (DAug+AM, Hendrycks et al., 2020a)[1]. For the ResNeXt model, we also include a version that was trained on 3.5 billion weakly labeled images (IG-3.5B, Mahajan et al., 2018). Finally, for EfficientNet-L2 we select the current state of the art on IN-C which was trained on 300 million images from JFT-300M (Chollet, 2017; Hinton et al., 2014) using a noisy student-teacher protocol (Xie et al., 2020a). We validate the IN and IN-C performance of all considered models and match the originally reported scores (Schneider et al., 2020). For EfficientNet-L2, we match IN top-1 accuracy up to 0.1% points, and IN-C up to 0.6% mCE.

**Models for CIFAR10/MNIST-scale datasets.** For CIFAR10-C experiments, we use two WideResNets (WRN, Zagoruyko & Komodakis, 2016): the first one is trained on CIFAR10 and has a depth of 28 and a width of 10 and the second one is trained with AugMix (Hendrycks et al., 2020b) and has a depth of 40 and a width of 2. The remaining small-scale models are trained with unsupervised domain adaptation (UDA) methods. We propose to regard any UDA method which requires joint training with source and target data as a pre-training step, similar to regular pre-training on IN, and use self-learning on top of the final checkpoint. We consider two popular UDA methods: self-supervised domain adaptation (UDA-SS; Sun et al., 2019a) and Domain-Adversarial Training of Neural Networks (DANN; Ganin et al., 2016). In UDA-SS, the authors seek to align the representations of both domains by performing an auxiliary self-supervised task on both domains simultaneously. In all UDA-SS experiments, we use a WideResNet with a depth of 26 and a width of 16. In DANN, the authors learn a domain-invariant embedding by optimizing a minimax objective. For all DANN experiments except for MNIST→MNIST-M, we use the same WRN architecture as above. For the MNIST→MNIST-M experiment, the training with the larger model diverged and we used a smaller WideResNet version with a width of 2. We note that DANN training involves optimizing a minimax objective and is generally harder to tune.

## 4 RESULTS: SELF-LEARNING UNIVERSALLY IMPROVES MODELS

Self-learning is a powerful learning scheme, and in the following section we show that it allows to perform test-time adaptation on robustified models, models obtained with large-scale pre-training, as well as domain adapted models across a wide range of datasets and distribution shifts. Our main results on large-scale and small-scale datasets are shown in Tables 1 and 2, respectively. These

---

[1]see leaderboard at `github.com/hendrycks/robustness`

summary tables show final results, and all experiments use the hyperparameters we determined separately on the dev set.

**Table 1: Self-learning successfully adapts ImageNet-scale models across different model architectures on IN-C, IN-A and IN-R.** We adapt the vanilla ResNet50, ResNeXt101 and DenseNet161 models to IN-C and decrease the mCE by over 19 percent points in all models. Further, self-learning works for models irrespective of their size: Self-learning substantially improves the performance of the ResNet50 and the ResNext101 trained with DAug+AM, on IN-C by 11.9 and 9.7 percent points, respectively. Finally, we further improve the current state-of-the-art model on IN-C—the EfficientNet-L2 Noisy Student model—and report a new state-of-the-art result of 22% mCE (which corresponds to a top1 error of 17.1%) on this benchmark with test-time adaptation (compared to 28% mCE without adaptation).

| | number of parameters | w/o adapt | w/ adapt RPL | $\Delta$ |
|---|---|---|---|---|
| mCE [%] on IN-C test ($\searrow$) | | | | |
| ResNet50 vanilla (He et al., 2016b) | $2.6 \times 10^7$ | 76.7 | 50.5 | (-26.2) |
| ResNet50 DAug+AM (Hendrycks et al., 2020a) | $2.6 \times 10^7$ | 53.6 | 41.7 | (-11.9) |
| DenseNet161 vanilla (Huang et al., 2017) | $2.8 \times 10^7$ | 66.4 | 47.0 | (-19.4) |
| ResNeXt101$_{32 \times 8d}$ vanilla (Xie et al., 2017) | $8.8 \times 10^7$ | 66.6 | 43.2 | (-23.4) |
| ResNeXt101$_{32 \times 8d}$ DAug+AM (Hendrycks et al., 2020a) | $8.8 \times 10^7$ | 44.5 | 34.8 | (-9.7) |
| ResNeXt101$_{32 \times 8d}$ IG-3.5B (Mahajan et al., 2018) | $8.8 \times 10^7$ | 51.7 | 40.9 | (-10.8) |
| EfficientNet-L2 Noisy Student (Xie et al., 2020a) | $4.8 \times 10^8$ | 28.3 | **22.0** | (-6.3) |
| top1 error [%] on IN-R ($\searrow$) | | | | |
| ResNet50 vanilla (He et al., 2016b) | $2.6 \times 10^7$ | 63.8 | 54.1 | (-9.7) |
| EfficientNet-L2 Noisy Student (Xie et al., 2020a) | $4.8 \times 10^8$ | 23.5 | **17.4** | (-6.1) |
| top1 error [%] on ImageNet-A ($\searrow$) | | | | |
| EfficientNet-L2 Noisy Student (Xie et al., 2020a) | $4.8 \times 10^8$ | 16.5 | **14.8** | (-1.7) |

Self-learning is not limited to the distribution shifts in IN-C like compression artefacts or blur. On IN-R, a dataset with renditions, self-learning improves both the vanilla ResNet50 and the EfficientNet-L2 model, the latter of which improves from 23.5% to a new state-of-the art of 17.4% top-1 error. For a vanilla ResNet50, we improve the top-1 error from 63.8% (Hendrycks et al., 2020a) to 54.1%. On IN-A, adapting the EfficientNet-L2 model using self-learning decreases the top-1 error from 16.5% (Xie et al., 2020a) to 14.8% top-1 error, again constituting a new state of the art with test-time adaptation on this dataset.

**Table 2: Self-learning improves robustified and domain adapted models on small-scale datasets.** We test common domain adaptation techniques like DANN (Ganin et al., 2016) and UDA-SS (Sun et al., 2019a), and show that self-learning is effective at further tuning such models to the target domain. We suggest to view unsupervised source/target domain adaptation as a step comparable to pre-training under corruptions, rather than an adaptation technique specifically tuned to the target set—indeed, we can achieve error rates using, e.g., DANN + target adaptation previously only possible with source/target based pseudo-labeling, across different common domain adaptation benchmarks. Self-learning also decreases the error on CIFAR10-C of the Wide ResNet model trained with AugMix (AM, Hendrycks et al., 2020b) and reaches a new state of the art on CIFAR10-C of 8.5% top1 error with test-time adaptation. †denotes preliminary results on CIFAR-C dev only, due to instabilities in training the adversarial network in DANN.

| | number of parameters | w/o adapt | w/ adapt ENT | $\Delta$ |
|---|---|---|---|---|
| top1 error [%] on CIFAR10-C ($\searrow$) | | | | |
| WRN-28-10 vanilla (Zagoruyko & Komodakis, 2016) | $3.6 \times 10^7$ | 26.5 | 13.3 | (-13.2) |
| WRN-40-2 AM (Hendrycks et al., 2020b) | $2.2 \times 10^6$ | 11.2 | 8.5 | (-2.7) |
| WRN-26-16 UDA-SS (Sun et al., 2019a) | $9.3 \times 10^7$ | 27.7 | 16.7 | (-11.0) |
| WRN-26-16 DANN (Ganin et al., 2016) | $9.3 \times 10^7$ | †29.7 | †28.5 | (-1.2) |
| UDA CIFAR10→STL10, top1 error on target [%]($\searrow$) | | | | |
| WRN-26-16 UDA-SS (Sun et al., 2019a) | $9.3 \times 10^7$ | 28.7 | 21.8 | (-6.9) |
| WRN-26-16 DANN (Ganin et al., 2016) | $9.3 \times 10^7$ | 25.0 | 23.9 | (-1.1) |
| UDA MNIST→MNIST-M, top1 error on target [%]($\searrow$) | | | | |
| WRN-26-16 UDA-SS (Sun et al., 2019a) | $9.3 \times 10^7$ | 4.8 | 2.0 | (-2.8) |
| WRN-26-2 DANN (Ganin et al., 2016) | $1.5 \times 10^6$ | 11.4 | 5.1 | (-6.3) |

**Table 3: Self-learning also improves large pre-trained models.** Unlike BatchNorm adaptation (Schneider et al., 2020), we show that self-learning transfers well to models pre-trained on a large amount of unlabeled data: self-learning decreases the mCE on IN-C of the ResNeXt101 trained on 3.5 billion weakly labeled samples (IG-3.5B, Mahajan et al., 2018) from 51.7% to 40.9%.

| mCE on IN-C test [%] ($\searrow$) | no adaptation | BN adaptation | self-learning |
|---|---|---|---|
| ResNeXt101$_{32\times8d}$ vanilla | 66.6 | 56.8 | 43.2 |
| ResNeXt101$_{32\times8d}$ IG-3.5B | 51.7 | 51.8 | **40.9** |

**Table 4: Self-learning outperforms previously published test-time adaptation approaches on IN-C.** The robustness benchmark IN-C has so far mostly been regarded in the ad-hoc evaluation setting as discussed in our introduction. Thus, there are only few published methods that report numbers for test-time adaptation: BatchNorm adaptation (Schneider et al., 2020), Test-Time Training (TTT, Sun et al., 2019b), and TENT (Wang et al., 2020). In particular, note that TTT requires a special loss function at training time, while our approach is agnostic to the pre-training phase. Our self-training results outperforms all three baselines (also after tuning TENT with our full experimental protocol):

| mCE on IN-C test [%] ($\searrow$) | w/o adapt | BN adapt | TENT (ours) | self-learning |
|---|---|---|---|---|
| ResNet50 vanilla | 76.7 | 62.2 | 53.5 (51.6) | **50.5** |
| top1 error [%] on IN-C, sev. 5 ($\searrow$) | w/o adapt | BN adapt | TTT | self-learning |
| ResNet18 vanilla | 85.4 | 72.2 | 66.3 | **61.9** |

**Table 5: Self-supervised methods based on self-learning allow out-of-the-box test-time adaptation.** The recently published DINO method (Caron et al., 2021) is another variant of self-supervised learning that has proven to be effective for unsupervised representation learning. At the core, the method uses soft pseudo-labeling. Here, we test whether a model trained with DINO on the source dataset can be test-time adapted on IN-C using DINO to further improve out-of-distribution performance. Since the used model is a vision transformer model, we test different choices of adaptation parameters and find considerable performance improvements in all cases, yielding an mCE of 43.5%mCE at a parameter count comparable to a ResNet50 model. For adapting the affine layers, we follow Houlsby et al. (2019):

| mCE on IN-C test [%] ($\searrow$) | w/o adapt | w/ adapt affine layers | w/ adapt bottleneck layers | w/ adapt lin. layers | w/ adapt all weights |
|---|---|---|---|---|---|
| ViT-S/16 | 62.3 | 51.8 | 46.8 | 45.2 | **43.5** |

## 5 UNDERSTANDING TEST-TIME ADAPTATION WITH SELF-LEARNING

In the following section, we show ablations and interesting insights of using self-learning for test-time adaptation. If not specified otherwise, all ablations are run on the holdout corruptions of IN-C (our dev set) with a vanilla ResNet50.

**Table 6: Robust pseudo-labeling outperforms entropy minimization on large-scale datasets while the reverse is true on small-scale datasets.** We find that robust pseudo-labeling consistently improves over entropy minimization on IN-C, while entropy minimization performs better on smaller scale data (CIFAR10, STL10, MNIST). The finding highlights the importance of testing both algorithms on new datasets. The improvement is typically on the order of one percent point:

| mCE, IN-C dev | ResNet50 | ResNeXt-101 | EfficientNet-L2 | | top-1 err, CIFAR-C | WRN-40 |
|---|---|---|---|---|---|---|
| ENT | $50.0 \pm 0.04$ | 43.0 | 22.2 | | ENT | **8.5** |
| RPL | $\mathbf{48.9 \pm 0.02}$ | **42.0** | **21.3** | | RPL | 9.0 |

**Table 7: Robust pseudo-labeling allows usage of the full dataset without a threshold.** Classical hard labeling needs a confidence threshold (T) for best performance, thereby reducing the dataset size, while best performance for RPL is reached for full dataset training with a threshold T of 0.0:

| diff. self-learning methods | no adapt | soft PL | hard PL (T): 0.0 | **0.5** | 0.9 | RPL (T): **0.0** | 0.5 | 0.9 |
|---|---|---|---|---|---|---|---|---|
| mCE on IN-C dev [%] | 69.5 | 60.1 | 53.8 | **51.9** | 52.4 | **49.7** | 49.9 | 51.8 |

**Table 8: Short update intervals are crucial for fast adaptation.** Having established that RPL generally performs better than soft- and hard-labeling, we vary the update interval for the teacher. We find that instant updates are most effective. In entropy minimization, the update interval is instant per default.

| Update interval for RPL | w/o adapt | no update | epoch | instant |
|---|---|---|---|---|
| mCE on IN-C dev [%] | 69.5 | 54.0 | 49.7 | **49.2** |

**Table 9: Adaptation of only affine layers is important in CNNs.** On IN-C, adapting only the affine parameters after the normalization layers (i.e., the rescaling and shift parameters $\beta$ and $\gamma$) works better on a ResNet50 architecture than adapting all parameters or only the last layer. We indicate the number of adapted parameters in brackets.

| Adaptation mechanism | w/o adapt | last layer | full model | affine |
|---|---|---|---|---|
| mCE on IN-C dev [%] | 69.5 [0] | 60.2 [2M] | 51.5 [22.6M] | **48.9** [5.3k] |

Note that for Vision Transformers, full model adaptation works better than affine adaptation (see Table 5). We also noticed that on convolutional models with a smaller parameter count like ResNet18, full model adaptation is possible.

**Hyperparameters obtained on corruption datasets transfer well to real world datasets.** When evaluating models, we select the hyperparameters discussed above (the learning rate and the epoch used for early stopping are the most critical ones) on the holdout set of IN-C. We note that this technique transfers well to IN-R, -A and -D, highlighting the practical value of corruption robustness datasets for adapting models on real distribution shifts.

On IN-D, we performed a control experiment where we selected hyperparameters with leave-one-out cross validation—this selection scheme actually performed worse than IN-C parameter selection (see Appendix D.1).

## 6    ADAPTING MODELS ON A WIDER RANGE OF DISTRIBUTION SHIFTS REVEALS LIMITATIONS OF ROBUSTIFICATION AND ADAPTATION METHODS

Robustness datasets on ImageNet-scale have so far been limited to a few selected domains (image corruptions in IN-C, image renditions in IN-R, difficult images for ResNet50 classifiers in IN-A). In order to test our approach on a wider range of complex distribution shifts, we re-purpose the dataset from the Visual Domain Adaptation Challenge 2019 (DomainNet, Saenko et al., 2019) as an additional robustness benchmark. This dataset comes with six image styles: Clipart, Real, Infograph, Painting, Quickdraw and Sketch. It has 345 classes in total, of which 164 overlap with IN. To benchmark robustness of IN trained models out of the box, we filter out the classes that cannot be mapped to IN and refer to the smaller version of DomainNet as ImageNet-D (IN-D). We map 463 classes in IN to these 164 IN-D classes, e.g., for an image from the "bird" class in IN-D, we accept all 39 bird classes in IN as valid predictions. We show example images from IN-D in Table 10. The detailed evaluation protocol along with justifications for our design choices and additional analysis are outlined in Appendix D.

The benefit of IN-D over DomainNet is the re-mapping to ImageNet classes which allows robustness researchers to easily benchmark on this dataset, without the need of re-training a model (as common in UDA). To test whether self-learning is helpful for more complex distribution shifts, we adapt a vanilla ResNet50, several robust IN-C models and the EfficientNet-L2 Noisy Student model on IN-D. We use the same hyperparameters we obtained on IN-C dev for all our IN-D experiments. We show our main results in Table 10.

**More robust models perform better on IN-D.** Comparing the performance of the vanilla ResNet50 model to its robust DAug+AM variant, we find that the DAug+AM model performs better on all domains, with the most significant gains on the "Clipart", "Painting" and "Sketch" domains. We show detailed results for all domains and all tested models in Appendix D.2, along with results on IN-C and IN-R for comparison. We find that the best performing models on IN-D are also the

Table 10: Self-learning decreases the top1 error on some IN-D domains but increases it on others.

| domain | Real | | Painting | | Clipart | | Sketch | | Infograph | | Quickdraw | |
| adapt | w/o | w/ | w/o | w/ | w/o | w/ | w/o | w/ | w/o | w/ | w/o | w/ |
| model | | | | | | | | | | | | |
| EffNet-L2 Noisy Student | 29.2 | **27.9** | 42.7 | **40.9** | 45.0 | **37.9** | 56.4 | **51.5** | 77.9 | 94.3 | **98.4** | 99.4 |
| ResNet50 DAug+AM | 39.2 | 36.5 | 58.7 | 53.4 | 68.4 | 57.0 | 75.2 | 61.3 | 88.1 | 83.2 | 98.2 | 99.1 |
| ResNet50 vanilla | 40.1 | 37.3 | 65.1 | 57.8 | 76.0 | 63.6 | 82.0 | 73.0 | 89.6 | 85.1 | 99.2 | 99.8 |

strongest ones on IN-C and IN-R which indicates good generalization capabilities of the techniques combined for these models, given the large differences between the three considered datasets. However, even the best models perform 20 to 30 percentage points worse on IN-D compared to their performance on IN-C or IN-R, indicating that IN-D might be a more challenging benchmark.

**All models struggle with some domains of IN-D.** The EfficientNet-L2 Noisy Student model obtains the best results on most domains. However, we note that the overall error rates are surprisingly high compared to the model's strong performance on the other considered datasets (IN-A: 14.8% top-1 error, IN-R: 17.4% top-1 error, IN-C: 22.0% mCE). Even on the "Real" domain closest to clean IN where the EfficientNet-L2 model has a top-1 error of 11.6%, the model only reaches a top-1 error of 29.2%. Self-learning decreases the top1 error on all domains except for "Infograph" and "Quickdraw". We note that both domains have very high error rates from the beginning and thus hypothesize that the produced pseudo-labels are of low quality.

**Error analysis on IN-D.** We investigate the errors a ResNet50 model makes on IN-D by analyzing the most frequently predicted classes for different domains to reveal systematic errors indicative of the encountered distribution shifts. We find most errors interpretable: the classifier assigns the label "comic book" to images from the "Clipart" or "Painting" domains, "website" to images from the "Infograph" domain, and "envelope" to images from the "Sketch" domain. Thus, the classifier predicts the domain rather than the class. We find no systematic errors on the "Real" domain which is expected since this domain should be similar to IN. Detailed results on the top-3 most frequently predicted classes for different domains can be found in Fig. 9, Appendix D.4.

**IN-D should be used as an additional robustness benchmark.** While the error rates on IN-C, -R and -A are at a well-acceptable level for our largest EfficientNet-L2 model after adaptation, IN-D performance is consistently worse for all models. We propose to move from isolated benchmark settings like IN-R (single domain) to benchmarks more common in domain adaptation (like DomainNet) and make IN-D publicly available as an easy to use dataset for this purpose.

**Additional experiments and limitations.** We discuss additional proof-of-concept implementations on the WILDS benchmark (Koh et al., 2021), BigTransfer (BiT; Chen et al., 2020a) models and on self-learning based UDA models in Appendix E. On WILDS, self-learning is effective for the Camelyon17 task with a systematic shift between train, validation and test sets (each set is comprised of different hospitals), while self-learning fails to improve on tasks with mixed domains.

## 7 A SIMPLE MODEL OF STABILITY IN SELF-LEARNING

We observed that different self-learning schemes are optimal for small-scale vs. large-scale datasets and varying amount of classes. We reconsider the used loss functions, and unify them into

$$\ell(\mathbf{x}) = -\sum_j \sigma_j \left( \frac{\mathbf{f}^t(\mathbf{x})}{\tau_t} \right) \log \left( \sigma_j \left( \frac{\mathbf{f}^s(\mathbf{x})}{\tau_s} \right) \right),$$

$$\mathbf{f}^t(\mathbf{x}) = \begin{cases} \mathbf{f}(\mathbf{x}), & \text{entropy minimization} \\ \text{sg}(\mathbf{f}(\mathbf{x})), & \text{pseudo-labeling.} \end{cases}$$

(5)

We introduced student and teacher temperature $\tau_s$ and $\tau_t$ as parameters in the softmax function and the stop gradient operation sg. Caron et al. (2021) fixed $\tau_s$ and varied $\tau_t$ during training,

and empirically found an upper bound for $\tau_t$ above which the training was no longer stable. To better understand such behavior, we study the learning dynamics of the loss function in equation 5 theoretically in a simple two-datapoints, two-classes model with linear student and teacher networks $\mathbf{f}^s(\mathbf{x}) = \mathbf{x}^\top \mathbf{w}^s$ and $\mathbf{f}^t(\mathbf{x}) = \mathbf{x}^\top \mathbf{w}^t$ defined in Appendix A.1. Gradient descent with stop gradient corresponds to hard pseudo-labeling in the limit $\tau_t \to 0$ and to soft pseudo-labeling when $\tau_s = \tau_t = 1$. Gradient descent without stop gradient, i.e., setting $\mathbf{w}^s = \mathbf{w}^t = \mathbf{w}$ corresponds to entropy minimization. We obtain the following result:

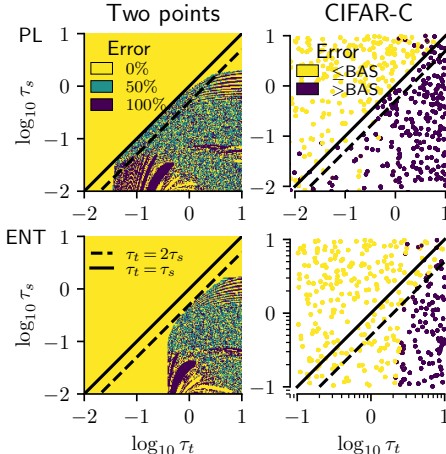

**Proposition 1** (Collapse in the two-point model). *The student and teacher networks $\mathbf{w}_s$ and $\mathbf{w}_t$ trained with stop gradient does not collapse to the trivial representation $\forall \mathbf{x} : \mathbf{x}^\top \mathbf{w}^s = 0, \mathbf{x}^\top \mathbf{w}^t = 0$ if $\tau_s > \tau_t$. The network $\mathbf{w}$ trained without stop gradient does not collapse if $\tau_s > \tau_t/2$. Proof. see § A.2.* □

We validate the proposition on a simulated two datapoint toy dataset, as well as on the CIFAR-C dataset and outline the results in Figure 2. In general, the size and location of the region where collapse is observed in the simulated model also depends on the initial conditions, the learning rate and the optimization procedure. An in depth discussion, as well as additional simulations are given in the Appendix. In practice, the result suggests that *student temperatures should exceed the teacher temperatures for pseudo-labeling*, and *student temperatures should exceed half the teacher temperature for entropy minimization*.

Entropy minimization with standard temperatures ($\tau_s = \tau_t = 1$) and hard pseudo-labeling ($\tau_t \to 0$) are hence stable. The two-point learning dynamics vanish for soft pseudo-labeling with $\tau_s = \tau_t$,

Figure 2: For the two point model, we show error and for the CIFAR10-C simulation, we show improvement (yellow) vs. degradation (purple) over the non-adapted baseline (BAS). An important convergence criterion for pseudo-labeling (top row) and entropy minimization (bottom row) is the ratio of student and teacher temperatures; it lies at $\tau_s = \tau_t$ for PL, and $2\tau_s = \tau_t$ for ENT. Despite the simplicity of the two-point model, the general convergence regions transfer to CIFAR10-C.

suggesting that one would have to analyze a more complex model with more data points. While this does not directly imply that the learning is unstable at this point, we empirically observe that both entropy minimization and hard labeling outperform soft-labeling in practice.

# 8 CONCLUSION

We evaluated and analysed how self-learning, an essential component in many unsupervised domain adaptation and self-supervised pre-training techniques, can be applied for adaptation to both small and large-scale image recognition problems common in robustness research. We demonstrated new state-of-the-art adaptation results with the EfficientNet-L2 model on the benchmarks ImageNet-C, -R, and -A, and introduced a new benchmark dataset (ImageNet-D) which remains challenging even after adaptation. Our theoretical analysis shows the influence of the temperature parameter in the self-learning loss function on the training stability and provides guidelines how to choose a suitable value. Self-learning universally improves test-time performance under diverse, but systematic distribution shifts irrespective of the architecture or pre-training method. We hope that our work encourages both researchers and practitioners to *use self-learning if their data distribution shifts.*

**Reproducibility Statement** We attempted to make our work as reproducible as possible: We mostly used pre-trained models which are publicly available and we denoted the URL addresses of all used checkpoints; for the checkpoints that were necessary to retrain, we report the Github directories with the source code and used an official or verified reference implementation when available. We report all used hyperparameters in the Appendix and will release our code upon acceptance of the paper.

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

# A A TWO-POINT MODEL OF SELF-LEARNING

## A.1 DEFINITION OF THE TWO-POINT MODEL

To understand the learning dynamics and properties of different loss functions and their hyperparameters, we propose a simple model of self-learning, both for entropy minimization and pseudo-labeling.

A student network $\mathbf{w}^s \in \mathbb{R}^d$ and a teacher network $\mathbf{w}^t \in \mathbb{R}^d$ are trained on $N$ data points $\{\mathbf{x}_i\}_{i=1}^N$ with the cross-entropy loss function $\mathcal{L}$ defined as

$$\mathcal{L} = -\sum_{i=1}^N \ell(\mathbf{x}_i) = -\sum_{i=1}^N \left(\sigma_t(\mathbf{x}_i^\top \mathbf{w}^t) \log \sigma_s(\mathbf{x}_i^\top \mathbf{w}^s) + \sigma_t(-\mathbf{x}_i^\top \mathbf{w}^t) \log \sigma_s(-\mathbf{x}_i^\top \mathbf{w}^s)\right), \tag{6}$$
$$\text{where } \sigma_t(z) = \frac{1}{1 + e^{-z/\tau_t}} \text{ and } \sigma_s(z) = \frac{1}{1 + e^{-z/\tau_s}}.$$

Here $\tau_s$ and $\tau_t$ denote the student and teacher temperature parameters. With stop gradient, student and teacher evolve in time according to

$$\dot{\mathbf{w}}^s = -\nabla_{\mathbf{w}^s}\mathcal{L}\left(\mathbf{w}^s, \mathbf{w}^t\right), \quad \dot{\mathbf{w}}^t = \alpha(\mathbf{w}^s - \mathbf{w}^t), \tag{7}$$

where $\alpha$ is the learning rate of the teacher. Without stop gradient, student and teacher are set equal to each other, and they evolve as

$$\dot{\mathbf{w}} = -\nabla_{\mathbf{w}}\mathcal{L}(\mathbf{w}), \text{ where } \mathbf{w}^s = \mathbf{w}^t = \mathbf{w}. \tag{8}$$

We restrict the theoretical analysis to the time evolution of the components of $\mathbf{w}^{s,t}$ in direction of two data points $\mathbf{x}_k$ and $\mathbf{x}_l$, $y_k^{s,t} \equiv \mathbf{x}_k^\top \mathbf{w}^{s,t}$ and $y_l^{s,t} \equiv \mathbf{x}_l^\top \mathbf{w}^{s,t}$. All other components $y_i^{s,t}$ with $i \neq k, l$ are neglected to reduce the dimensionality of the equation system. It turns out that the resulting model captures the neural network dynamics quite well despite the drastic simplification of taking only two data points into account (see Figure 2).

$$\text{with stop gradient: } \dot{y}_k^s = -\mathbf{x}_k^\top \nabla_{\mathbf{w}^s}\left(\ell(\mathbf{x}_k) + \ell(\mathbf{x}_l)\right), \quad \dot{y}_l^s = -\mathbf{x}_l^\top \nabla_{\mathbf{w}^s}\left(\ell(\mathbf{x}_k) + \ell(\mathbf{x}_l)\right),$$
$$\dot{y}_k^t = \alpha(y_k^t - y_k^s), \quad \dot{y}_l^t = \alpha(y_l^t - y_l^s), \tag{9}$$
$$\text{without stop gradient: } \dot{y}_k = -\mathbf{x}_k^\top \nabla_{\mathbf{w}}\left(\ell(\mathbf{x}_k) + \ell(\mathbf{x}_l)\right), \quad \dot{y}_l = -\mathbf{x}_l^\top \nabla_{\mathbf{w}}\left(\ell(\mathbf{x}_k) + \ell(\mathbf{x}_l)\right).$$

## A.2 PROOF OF PROPOSITION 1

**Learning dynamics with stop gradient.** Computing the stop gradient evolution defined in equation 7 explicitly yields

$$\dot{\mathbf{w}}^s = -\nabla_{\mathbf{w}^s}\mathcal{L} = \frac{1}{\tau_s}\sum_{i=1}^N \left(\sigma_t(\mathbf{x}_i^\top \mathbf{w}^t)\sigma_s(-\mathbf{x}_i^\top \mathbf{w}^s) - \sigma_t(-\mathbf{x}_i^\top \mathbf{w}^t)\sigma_s(\mathbf{x}_i^\top \mathbf{w}^s)\right)\mathbf{x}_i \tag{10}$$
$$\dot{\mathbf{w}}^t = \alpha(\mathbf{w}^s - \mathbf{w}^t)$$

The second equality uses the well-known derivative of the sigmoid function, $\partial_z \sigma(z) = \sigma(z)\sigma(-z)$.

The equation system of $2d$ nonlinear, coupled ODEs for $\mathbf{w}^s \in \mathbb{R}^d$ and $\mathbf{w}^t \in \mathbb{R}^d$ in equation 10 is analytically difficult to analyze. Instead of studying the ODEs directly, we act on them with the data points $\mathbf{x}_k^\top$, $k = 1, \ldots, N$, and investigate the dynamics of the components $\mathbf{x}_k^\top \mathbf{w}^{s,t} \equiv y_k^{s,t}$:

$$\dot{y}_k^s = \frac{1}{\tau_s}\sum_{i=1}^N \left(\mathbf{x}_i^\top \mathbf{x}_k\right)\left(\sigma_t(y_i^t)\sigma_s(-y_i^s) - \sigma_t(-y_i^t)\sigma_s(y_i^s)\right) \tag{11}$$
$$\dot{y}_k^t = \alpha(y_k^s - y_k^t).$$

The learning rate of each mode $y_k^s$ is scaled by $(\mathbf{x}_k^\top \mathbf{x}_i)$ which is much larger for $i = k$ than for $i \neq k$ in high-dimensional spaces. In the two-point approximation, we consider only the two (in absolute

value) largest terms $i = k, l$ for a given $k$ in the sum in equation 11. Any changes that $y_k^{s,t}(t)$ and $y_l^{s,t}(t)$ might induce in other modes $y_i^{s,t}(t)$ are neglected, and so we are left with only four ODEs:

$$
\begin{aligned}
\dot{y}_k^s =& \frac{1}{\tau_s} \|\mathbf{x}_k\|^2 \left( \sigma_t(y_k^t)\sigma_s(-y_k^s) - \sigma_t(-y_k^t)\sigma_s(y_k^s) \right) \\
&+ \frac{1}{\tau_s} (\mathbf{x}_k^\top \mathbf{x}_l) \left( \sigma_t(y_l^t)\sigma_s(-y_l^s) - \sigma_t(-y_l^t)\sigma_s(y_l^s) \right), \\
\dot{y}_l^s =& \frac{1}{\tau_s} \|\mathbf{x}_l\|^2 \left( \sigma_t(y_l^t)\sigma_s(-y_l^s) - \sigma_t(-y_l^t)\sigma_s(y_l^s) \right) \\
&+ \frac{1}{\tau_s} (\mathbf{x}_k^\top \mathbf{x}_l) \left( \sigma_t(y_k^t)\sigma_s(-y_k^s) - \sigma_t(-y_k^t)\sigma_s(y_k^s) \right) \\
\dot{y}_k^t =& \alpha(y_k^s - y_k^t), \; \dot{y}_l^t = \alpha(y_l^s - y_l^t).
\end{aligned}
\tag{12}
$$

The fixed points of equation 12 satisfy

$$
\dot{y}_k^s = \dot{y}_l^s = \dot{y}_k^t = \dot{y}_l^t = 0.
\tag{13}
$$

For $\alpha > 0$, requiring $\dot{y}_k^t = \dot{y}_l^t = 0$ implies that $y_k^s = y_k^t$ and $y_l^s = y_l^t$. For $\tau_s = \tau_t$, the two remaining equations $\dot{y}_k^s = \dot{y}_l^s = 0$ vanish automatically so that there are no non-trivial two-point learning dynamics. For $\tau_s \neq \tau_t$, there is a fixed point at $y_k^{s,t} = y_l^{s,t} = 0$ since at this point, each bracket in equation 12 vanishes individually:

$$
\sigma_t(y_{k,l})\sigma_s(-y_{k,l}) - \sigma_s(-y_{k,l})\sigma_t(y_{k,l}) \Big|_{y_{k,l}=0} = \frac{1}{4} - \frac{1}{4} = 0.
\tag{14}
$$

At the fixed point $y_k^{s,t} = y_l^{s,t} = 0$, $\mathbf{w}^s$ and $\mathbf{w}^t$ are orthogonal to both $\mathbf{x}_k$ and $\mathbf{x}_l$ and hence classification fails. If this fixed point is stable, $\mathbf{w}^s$ and $\mathbf{w}^t$ will stay at the fixed point once they have reached it, i.e. the model collapses. The fixed point is stable when all eigenvalues of the Jacobian $J$ of the ODE system equation 12 evaluated at $y_k^{s,t} = y_l^{s,t} = 0$ are negative. This is the case whenever $\tau_s < \tau_t$:

$$
J \Big|_{y_k^{s,t}=y_l^{s,t}=0} = \begin{pmatrix} \frac{\|\mathbf{x}_k\|^2}{4}\left(\frac{1}{\tau_t}-\frac{1}{\tau_s}\right) & \frac{(\mathbf{x}_k^\top \mathbf{x}_l)}{4}\left(\frac{1}{\tau_t}-\frac{1}{\tau_s}\right) & 0 & 0 \\ \frac{(\mathbf{x}_k^\top \mathbf{x}_l)}{4}\left(\frac{1}{\tau_t}-\frac{1}{\tau_s}\right) & \frac{\|\mathbf{x}_l\|^2}{4}\left(\frac{1}{\tau_t}-\frac{1}{\tau_s}\right) & 0 & 0 \\ \alpha & 0 & -\alpha & 0 \\ 0 & \alpha & 0 & -\alpha \end{pmatrix},
$$

eigenvalues: $\lambda_1 = \lambda_2 = -\alpha < 0$,

$$
\lambda_{3,4} = \frac{1}{8}\left(\frac{1}{\tau_t}-\frac{1}{\tau_s}\right)\left( \pm \underbrace{\underbrace{\sqrt{\|\mathbf{x}_k\|^4 + \|\mathbf{x}_l\|^4 - 2\|\mathbf{x}_k\|^2\|\mathbf{x}_l\|^2 + 4(\mathbf{x}_k^\top \mathbf{x}_l)^2}}_{\leq \|\mathbf{x}_k\|^2 + \|\mathbf{x}_l\|^2} + \|\mathbf{x}_k\|^2 + \|\mathbf{x}_l\|^2}_{\geq 0 \text{ with equality if } \mathbf{x}_k = \pm \mathbf{x}_l} \right)
\tag{15}
$$

To sum up, training with stop gradient and $\tau_s > \tau_t$ avoids a collapse of the two-point model to the trivial representation $y_k^{s,t} = y_l^{s,t} = 0$ since the fixed point is not stable in this parameter regime.

**Learning dynamics without stop gradient**  Without stop gradient, we set $\mathbf{w}^t = \mathbf{w}^s \equiv \mathbf{w}$ which leads to an additional term in the gradient:

$$
\begin{aligned}
\dot{\mathbf{w}} = -\nabla_{\mathbf{w}}\mathcal{L} =& \frac{1}{\tau_s}\sum_{i=1}^N \left( \sigma_t(\mathbf{x}_i^\top \mathbf{w})\sigma_s(-\mathbf{x}_i^\top \mathbf{w}) - \sigma_t(-\mathbf{x}_i^\top \mathbf{w})\sigma_s(\mathbf{x}_i^\top \mathbf{w}) \right) \mathbf{x}_i \\
&+ \frac{1}{\tau_t}\sum_{i=1}^N \sigma_t(\mathbf{x}_i^\top \mathbf{w})\sigma_t(-\mathbf{x}_i^\top \mathbf{w}) \underbrace{\left( \log\sigma_s(\mathbf{x}_i^\top \mathbf{w}) - \log\sigma_s(-\mathbf{x}_i^\top \mathbf{w}) \right)}_{=\log\left((1+e^{y_i/\tau_s})/(1+e^{-y_i/\tau_s})\right)=y_i/\tau_s} \mathbf{x}_i.
\end{aligned}
\tag{16}
$$

As before, we focus on the evolution of the two components $y_k = \mathbf{w}^\top \mathbf{x}_k$ and $y_l = \mathbf{w}^\top \mathbf{x}_l$.

$$
\begin{aligned}
\dot{y}_k =& \|\mathbf{x}_k\|^2 \left( \frac{1}{\tau_s} \left( \sigma_t(y_k)\sigma_s(-y_k) - \sigma_t(-y_k)\sigma_s(y_k) \right) + \frac{1}{\tau_t}\sigma_t(y_k)\sigma_t(-y_k)y_k \right) \\
&+ (\mathbf{x}_k^\top \mathbf{x}_l) \left( \frac{1}{\tau_s} \left( \sigma_t(y_l)\sigma_s(-y_l) - \sigma_t(-y_l)\sigma_s(y_l) \right) + \frac{1}{\tau_s\tau_t}\sigma_t(y_l)\sigma_t(-y_l)y_l \right) \\
\dot{y}_l =& \|\mathbf{x}_l\|^2 \left( \frac{1}{\tau_s} \left( \sigma_t(y_l)\sigma_s(-y_l) - \sigma_t(-y_l)\sigma_s(y_l) \right) + \frac{1}{\tau_t}\sigma_t(y_l)\sigma_t(-y_l)y_l \right) \\
&+ (\mathbf{x}_k^\top \mathbf{x}_l) \left( \frac{1}{\tau_s} \left( \sigma_t(y_k)\sigma_s(-y_k) - \sigma_t(-y_k)\sigma_s(y_k) \right) + \frac{1}{\tau_s\tau_t}\sigma_t(y_k)\sigma_t(-y_k)y_k \right)
\end{aligned}
\tag{17}
$$

There is a fixed point at $y_k = y_l = 0$ where each bracket in equation 17 vanishes individually,

$$
\frac{1}{\tau_s} \left( \sigma_t(y_{k,l})\sigma_s(-y_{k,l}) - \sigma_t(-y_{k,l})\sigma_s(y_{k,l}) \right) + \frac{1}{\tau_s\tau_t}\sigma_t(y_{k,l})\sigma_t(-y_{k,l})y_{k,l} \Big|_{y_{k,l}} = 0.
\tag{18}
$$

The Jacobian of the ODE system in equation 17 and its eigenvalues evaluated at the fixed point are given by

$$
J \Big|_{y_k=y_l=0} = \begin{pmatrix} \frac{\|\mathbf{x}_k\|^2}{4\tau_s}\left( \frac{2}{\tau_t} - \frac{1}{\tau_s} \right) & \frac{(\mathbf{x}_k^\top \mathbf{x}_l)}{4\tau_s}\left( \frac{2}{\tau_t} - \frac{1}{\tau_s} \right) \\ \frac{(\mathbf{x}_k^\top \mathbf{x}_l)}{4\tau_s}\left( \frac{2}{\tau_t} - \frac{1}{\tau_s} \right) & \frac{\|\mathbf{x}_l\|^2}{4\tau_s}\left( \frac{2}{\tau_t} - \frac{1}{\tau_s} \right) \end{pmatrix}
$$

$$
\lambda_{1,2} = \frac{1}{8\tau_s}\left( \frac{2}{\tau_t} - \frac{1}{\tau_s} \right) \left( \pm \underbrace{\sqrt{\underbrace{\|\mathbf{x}_k\|^4 + \|\mathbf{x}_l\|^4 - 2\|\mathbf{x}_k\|^2\|\mathbf{x}_l\|^2 + 4(\mathbf{x}_k^\top \mathbf{x}_l)^2}_{\leq \|\mathbf{x}_k\|^2 + \|\mathbf{x}_l\|^2}} + \|\mathbf{x}_k\|^2 + \|\mathbf{x}_l\|^2}_{\geq 0 \text{ with equality if } \mathbf{x}_k = \pm\mathbf{x}_l} \right).
\tag{19}
$$

Hence the fixed point is unstable when $\tau_s > \tau_t/2$ and thus the model without stop gradient does not collapse onto $y_k = y_l = 0$ in this regime.

### A.3 SIMULATION OF THE TWO-POINT MODEL

For visualization purposes in the main paper, we set $\mathbf{w}^s = \mathbf{w}^t = [0.5, 0.5]^\top$ and train the model using instant gradient updates on the dataset with points $\mathbf{x}_1 = [1, 0]$ and $\mathbf{x}_2 = [0, -1]$ using SGD with learning rate 0.1 and momentum 0.9. We varied student and teacher temperatures on a log-scale with 250 points from $10^{-3}$ to 10. Qualitatively similar results can be obtained without momentum training, at higher learning rates (most likely due to the implicit learning rate scaling introduced by the momentum term).

Note that the temperature scales for observing the collapse effect depend on the learning rate, and the exact training strategy—lower learning rates can empirically prevent the model from collapsing and shift the convergence region. The result in Figure 2 will hence depend on the exact choice of learning rate (which is currently not considered in our continuous time evolution theory), while the predicted region without collapse is robust to details of the optimization.

To visualize the impact of different hyperparameters, we show variants of the two point model with different learning rates using gradient descent with (Figure 3) and without momentum (Figure 4), and with different start conditions (Figure 5), which all influence the regions where the model degrades, but not the stable regions predicted by our theory.

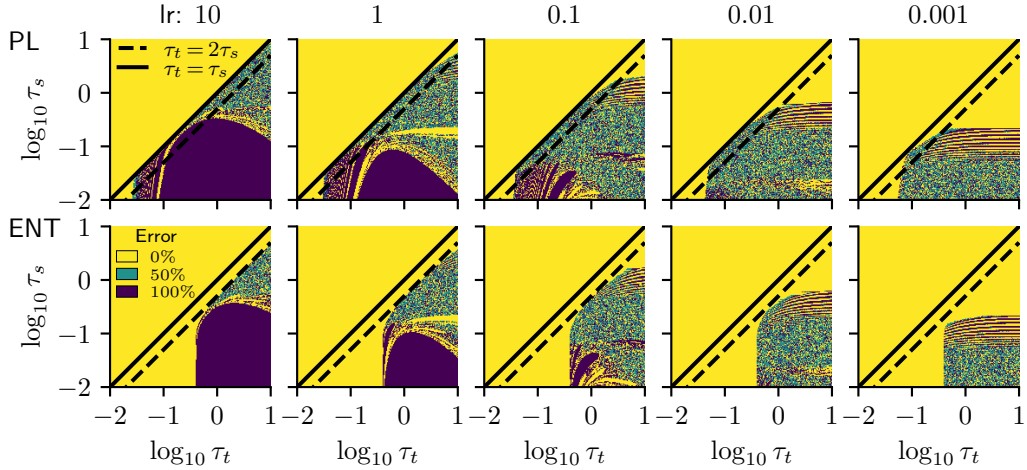

Figure 3: Entropy minimization (top) Training two point model with momentum 0.9 and different learning rates with initialization $\mathbf{w}^s = \mathbf{w}^t = [0.5, 0.5]^\top$.

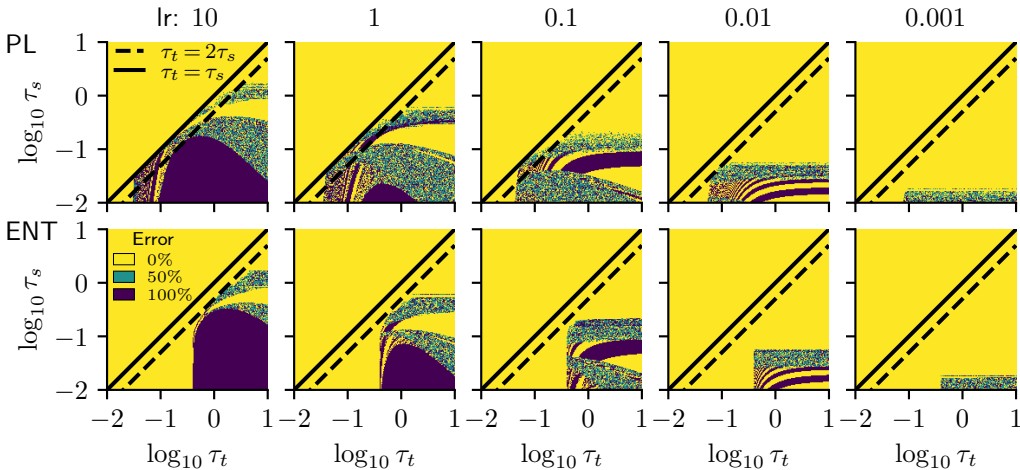

Figure 4: Training a two point model without momentum and different learning rates with initialization $\mathbf{w}^s = \mathbf{w}^t = [0.5, 0.5]^\top$. Note that especially for lower learning rates, longer training would increase the size of the collapsed region.

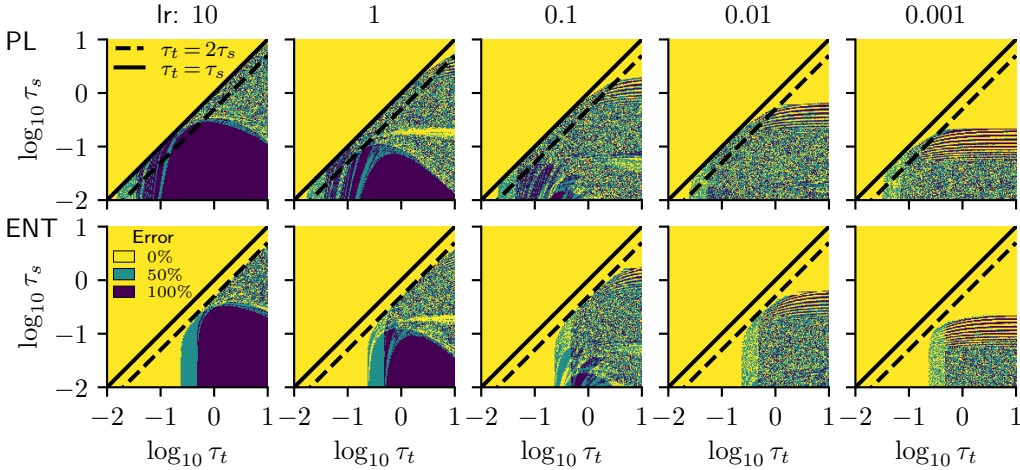

Figure 5: Training a two point model with momentum 0.9 and different learning rates with initialization $\mathbf{w}^s = \mathbf{w}^t = [0.6, 0.3]^\top$.

# B ADDITIONAL INFORMATION ON USED MODELS

## B.1 DETAILS ON ALL HYPERPARAMETERS WE TESTED FOR DIFFERENT MODELS

For all models except EfficientNet-L2, we adapt the batch norm statistics to the test domains following (Schneider et al., 2020). We do not expect significant gains for combining EfficientNet-L2 with batch norm adaptation: as demonstrated in (Schneider et al., 2020), models trained with large amounts of weakly labeled data do not seem to benefit from batch norm adaptation.

**ResNet50 models** We use a vanilla ResNet50 model and compare soft- and hard-labeling against entropy minimization and robust pseudo-labeling. To find optimal hyperparameters for all methods, we perform an extensive evaluation and test (i) three different adaptation mechanisms (ii) several learning rates $1.0 \times 10^{-4}$, $1.0 \times 10^{-3}$, $1.0 \times 10^{-2}$ and $5.0 \times 10^{-2}$, (iii) the number of training epochs and (iv) updating the teacher after each epoch or each iteration. For all experiments, we use a batch size of 128. The hyperparameter search is performed on IN-C dev. We then use the optimal hyperparameters to evaluate the methods on the IN-C test set.

**ResNeXt101 models** The ResNeXt101 model is considerably larger than the ResNet50 model and we therefore limit the number of ablation studies we perform for this architecture. Besides a baseline, we include a state-of-the-art robust version trained with DeepAugment+Augmix (DAug+AM, Hendrycks et al., 2020a) and a version that was trained on 3.5 billion weakly labeled images (IG-3.5B, Mahajan et al., 2018). We only test the two leading methods on the ResNeXt101 models (ENT and RPL). We vary the learning rate in same interval as for the ResNet50 model but scale it down linearly to account for the smaller batch size of 32. We only train the affine batch normalization parameters because adapting only these parameters leads to the best results on ResNet50 and is much more resource efficient than adapting all model parameters. Again, the hyperparameter search is performed only on the development corruptions of IN-C. We then use the optimal hyperparameters to evaluate the methods on the IN-C test set.

**EfficientNet-L2 models** The current state of the art on IN, IN-C, IN-R and IN-A is an EfficientNet-L2 trained on 300 million images from JFT-300M (Chollet, 2017; Hinton et al., 2014) using a noisy student-teacher protocol (Xie et al., 2020a). We adapt this model for only one epoch due to resource constraints. During the hyperparameter search, we only evaluate three corruptions on the IN-C development set[2] and test the learning rates $4.6 \times 10^{-2}$, $4.6 \times 10^{-3}$, $4.6 \times 10^{-4}$ and $4.6 \times 10^{-5}$. We use the optimal hyperparameters to evaluate ENT and RPL on the full IN-C test set (with all severity levels).

**UDA-SS models** We trained the models using the scripts from the official code base at github. com/yueatsprograms/uda_release. We used the provided scripts for the cases: (a) source: CIFAR10, target: STL10 and (b) source: MNIST, target: MNIST-M. For the case (c) source: CIFAR10, target: CIFAR10-C, we used the hyperparameters from case (a) since this case seemed to be the closest match to the new setting. We think that the baseline performance of the UDA-SS models can be further improved with hyperparameter tuning.

**DANN models** To train models with the DANN-method, we used the PyTorch implementation of this paper at https://github.com/fungtion/DANN_py3. The code base only provides scripts and hyperparameters for the case (b) source: MNIST, target: MNIST-M. For the cases (a) and (c), we used the same optimizer and trained the model for 100 epochs. We think that the baseline performance of the DANN models can be further improved with hyperparameter tuning.

**Preprocessing** For IN, IN-R, IN-A and IN-D, we resize all images to $256 \times 256$ px and take the center $224 \times 224$ px crop. The IN-C images are already rescaled and cropped. We center and re-scale the color values with $\mu_{RGB} = [0.485, 0.456, 0.406]$ and $\sigma_{RGB} = [0.229, 0.224, 0.225]$. For the EfficientNet-L2, we follow the procedure in Xie et al. (2020a) and rescale all inputs to a resolution of $507 \times 507$ px and then center-crop them to $475 \times 475$ px.

---

[2]We compare the results of computing the dev set on the 1, 3 and 5 severities versus the 1, 2, 3, 4 and 5 severities on our ResNeXt101 model in the Supplementary material.

## B.2  Full list of used models

**ImageNet scale models**  ImageNet trained models (ResNet50, DenseNet161, ResNeXt) are taken directly from torchvision (Marcel & Rodriguez, 2010). The model variants trained with DeepAugment and AugMix augmentations (Hendrycks et al., 2020b;a) are taken from https://github.com/hendrycks/imagenet-r. The weakly-supervised ResNeXt101 model is taken from the PyTorch Hub. For EfficientNet (Tan & Le, 2019), we use the PyTorch re-implementation available at https://github.com/rwightman/gen-efficientnet-pytorch. This is a verified re-implementation of the original work by Xie et al. (2020a). We verify the performance on ImageNet, yielding a 88.23% top-1 accuracy and 98.546% top-5 accuracy which is within 0.2% points of the originally reported result (Xie et al., 2020a). On ImageNet-C, our reproduced baseline achieves 28.9% mCE vs. 28.3% mCE originally reported by Xie et al. (2020a). As noted in the re-implementation, this offset is possible due to minor differences in the pre-processing. It is possible that our adaptation results would improve further when applied on the original codebase by Xie *et al.*.

**Small scale models**  We train the UDA-SS models using the original code base at github.com/yueatsprograms/uda_release, with the hyperparameters given in the provided bash scripts. For our DANN experiments, we use the PyTorch implementation at github.com/fungtion/DANN_py3. We use the hyperparameters in the provided bash scripts.

The following Table 11 contains all models we evaluated on various datasets with references and links to the corresponding source code.

Table 11: Model checkpoints used for our experiments.

| Model | Source |
|---|---|
| WideResNet(28,10) (Croce et al., 2020) | https://github.com/RobustBench/robustbench/tree/master/robustbench |
| WideResNet(40,2)+AugMix (Croce et al., 2020) | https://github.com/RobustBench/robustbench/tree/master/robustbench |
| ResNet50 (He et al., 2016b) | https://github.com/pytorch/vision/tree/master/torchvision/models |
| ResNeXt101, 32×8d (He et al., 2016b) | https://github.com/pytorch/vision/tree/master/torchvision/models |
| DenseNet (Huang et al., 2017) | https://github.com/pytorch/vision/tree/master/torchvision/models |
| ResNeXt101, 32×8d (Xie et al., 2017) | https://pytorch.org/hub/facebookresearch_WSL-Images_resnext/ |
| ResNet50+DeepAugment+AugMix (Hendrycks et al., 2020a) | https://github.com/hendrycks/imagenet-r |
| ResNext101 (Hendrycks et al., 2020a) | https://github.com/hendrycks/imagenet-r |
| ResNext101 32×8d IG-3.5B (Mahajan et al., 2018) | https://github.com/facebookresearch/WSL-Images/blob/master/hubconf.py |
| Noisy Student EfficientNet-L2 (Xie et al., 2020a) | https://github.com/rwightman/gen-efficientnet-pytorch |
| ViT-S/16 (Caron et al., 2021) | https://github.com/facebookresearch/dino |

## C  DETAILED AND ADDITIONAL RESULTS ON IN-C

### C.1  DEFINITION OF THE MEAN CORRUPTION ERROR (MCE)

The established performance metric on IN-C is the mean Corruption Error (mCE), which is obtained by normalizing the model's top-1 errors with the top-1 errors of AlexNet across the C=15 test corruptions and S=5 severities:

$$\text{mCE(model)} = \frac{1}{C} \sum_{c=1}^{C} \frac{\sum_{s=1}^{S} \text{err}_{c,s}^{\text{model}}}{\sum_{s=1}^{S} \text{err}_{c,s}^{\text{AlexNet}}}. \tag{20}$$

The AlexNet errors used for normalization are shown in Table 12.

| Category | Corruption | top1 error |
|---|---|---|
| Noise | Gaussian Noise | 0.886428 |
| | Shot Noise | 0.894468 |
| | Impulse Noise | 0.922640 |
| Blur | Defocus Blur | 0.819880 |
| | Glass Blur | 0.826268 |
| | Motion Blur | 0.785948 |
| | Zoom Blur | 0.798360 |
| Weather | Snow | 0.866816 |
| | Frost | 0.826572 |
| | Fog | 0.819324 |
| | Brightness | 0.564592 |
| | Contrast | 0.853204 |
| Digital | Elastic Transform | 0.646056 |
| | Pixelate | 0.717840 |
| | JPEG Compression | 0.606500 |
| Hold-out Noise | Speckle Noise | 0.845388 |
| Hold-out Digital | Saturate | 0.658248 |
| Hold-out Blur | Gaussian Blur | 0.787108 |
| Hold-out Weather | Spatter | 0.717512 |

Table 12: AlexNet top1 errors on ImageNet-C

### C.2  DETAILED RESULTS FOR TUNING EPOCHS AND LEARNING RATES

We tune the learning rate for all models and the number of training epochs for all models except the EfficientNet-L2. In this section, we present detailed results for tuning these hyperparameters for all considered models. The best hyperparameters that we found in this analysis, are summarized in Table 17.

Table 13: mCE in % on the IN-C dev set for ENT and RPL for different numbers of training epochs when adapting the affine batch norm parameters of a ResNet50 model.

| criterion | ENT | | | RPL | | |
|---|---|---|---|---|---|---|
| lr | $10^{-4}$ | $10^{-3}$ | $10^{-2}$ | $10^{-4}$ | $10^{-3}$ | $10^{-2}$ |
| epoch | | | | | | |
| 0 | 60.2 | 60.2 | 60.2 | 60.2 | 60.2 | 60.2 |
| 1 | 54.3 | **50.0** | 72.5 | 57.4 | 51.1 | 52.5 |
| 2 | 52.4 | 50.9 | 96.5 | 55.8 | 49.6 | 57.4 |
| 3 | 51.5 | 51.0 | 112.9 | 54.6 | 49.2 | 64.2 |
| 4 | 51.0 | 52.4 | 124.1 | 53.7 | 49.0 | 71.0 |
| 5 | 50.7 | 53.5 | 131.2 | 52.9 | **48.9** | 76.3 |
| 6 | 50.7 | 53.5 | 131.2 | 52.9 | 48.9 | 76.3 |

Table 14: mCE (↘) in % on the IN-C dev set for different learning rates for EfficientNet-L2. We favor $q = 0.8$ over $q = 0.7$ due to slightly improved robustness to changes in the learning rate in the worst case error setting.

| lr (4.6 ×) | base | $10^{-3}$ | $10^{-4}$ | $10^{-5}$ | $10^{-6}$ |
|---|---|---|---|---|---|
| ENT | 25.5 | 87.8 | 25.3 | **22.2** | 24.1 |
| RPL$_{q=0.7}$ | 25.5 | 60.3 | **21.3** | 23.3 | n/a |
| RPL$_{q=0.8}$ | 25.5 | 58.2 | **21.4** | 23.4 | n/a |

Table 17: The best hyperparameters for all models that we found on IN-C. For all models, we fine-tune only the affine batch normalization parameters and use $q = 0.8$ for RPL. The small batchsize for the EfficientNet model is due to hardware limitations.

| Model | Method | Learning rate | batch size | number of epochs |
|---|---|---|---|---|
| vanilla ResNet50 | ENT | $1 \times 10^{-3}$ | 128 | 1 |
| vanilla ResNet50 | RPL | $1 \times 10^{-3}$ | 128 | 5 |
| vanilla ResNeXt101 | ENT | $2.5 \times 10^{-4}$ | 128 | 1 |
| vanilla ResNeXt101 | RPL | $2.5 \times 10^{-4}$ | 128 | 4 |
| IG-3.5B ResNeXt101 | ENT | $2.5 \times 10^{-4}$ | 128 | 4 |
| IG-3.5B ResNeXt101 | RPL | $2.5 \times 10^{-3}$ | 128 | 2 |
| DAug+AM ResNeXt101 | ENT | $2.5 \times 10^{-4}$ | 128 | 1 |
| DAug+AM ResNeXt101 | RPL | $2.5 \times 10^{-4}$ | 128 | 4 |
| EfficientNet-L2 | ENT | $4.6 \times 10^{-5}$ | 8 | 1 |
| EfficientNet-L2 | RPL | $4.6 \times 10^{-4}$ | 8 | 1 |

Table 15: mCE in % on IN-C dev for entropy minimization for different learning rates and training epochs for ResNeXt101. (div.=diverged)

| ENT lr $2.5 \times$ epoch | Baseline 1e-4 | 1e-3 | 5e-3 | IG-3.5B 1e-4 | 1e-3 | 5e-3 | DAug+AM 1e-4 | 1e-3 | 5e-3 |
|---|---|---|---|---|---|---|---|---|---|
| BASE | 53.6 | 53.6 | 53.6 | 47.4 | 47.4 | 47.4 | 37.4 | 37.4 | 37.4 |
| 1 | **43.0** | 92.2 | div. | 40.9 | 40.4 | 58.6 | **35.4** | 46.4 | div. |
| 2 | 44.8 | 118.4 | div. | 39.8 | 41.5 | 69.5 | 35.5 | 90.8 | div. |
| 3 | 45.4 | 131.9 | div. | 39.3 | 42.6 | 76.1 | 35.5 | 122.5 | div. |
| 4 | 46.7 | div. | div. | **39.1** | 44.2 | 84.3 | 35.6 | 133.8 | div. |

Table 16: mCE in % on IN-C dev for robust pseudo-labeling for different learning rates and training epochs for ResNeXt101. (div.=diverged)

| RPL lr $2.5 \times$ epoch | Baseline 1e-4 | 1e-3 | 5e-3 | IG-3.5B 1e-4 | 1e-3 | 5e-3 | DAug+AM 1e-4 | 1e-3 | 5e-3 |
|---|---|---|---|---|---|---|---|---|---|
| BASE | 53.6 | 53.6 | 53.6 | 47.4 | 47.4 | 47.4 | 37.4 | 37.4 | 37.4 |
| 1 | 43.4 | 51.3 | div. | 45.0 | 39.9 | 43.6 | 35.3 | 35.1 | 79.1 |
| 2 | 42.3 | 63.2 | div. | 43.4 | **39.3** | 48.2 | 34.9 | 35.6 | 121.2 |
| 3 | 42.0 | 72.6 | div. | 42.4 | 39.4 | 52.9 | 34.7 | 40.1 | 133.5 |
| 4 | **42.0** | 72.6 | div. | 42.4 | 39.4 | 52.9 | **34.7** | 40.1 | 133.5 |

## C.3 DETAILED RESULTS FOR ALL IN-C CORRUPTIONS

We outline detailed results for all corruptions and models in Table 18. Performance across the severities in the dataset is depicted in Figure 6. All detailed results presented here are obtained by following the model selection protocol outlined in the main text.

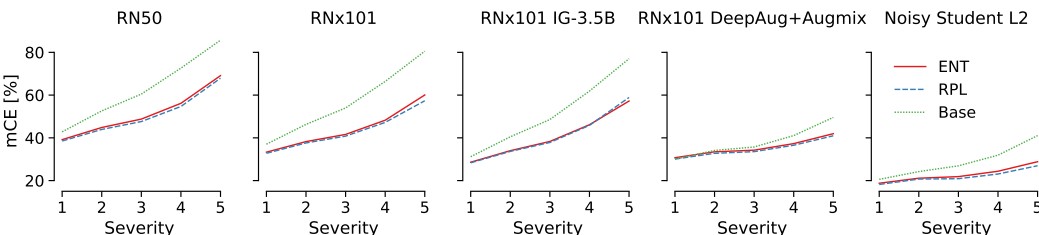

Figure 6: Severity-wise mean corruption error (normalized using the *average* AlexNet baseline error for each corruption) for ResNet50 (RN50), ResNext101 (RNx101) variants and the Noisy Student L2 model. Especially for more robust models (DeepAugment+Augmix and Noisy Student L2), most gains are obtained across higher severities 4 and 5. For weaker models, the baseline variant (Base) is additionally substantially improved for smaller corruptions.

Table 18: Detailed results for each corruption along with mean corruption error (mCE) as reported in Table 2 in the main paper. We show (unnormalized) top-1 error rate averaged across 15 test corruptions along with the mean corruption error (mCE: which is normalized). Hyperparameter selection for both ENT and RPL was carried out on the dev corruptions as outlined in the main text. Mismatch in baseline mCE for EfficientNet-L2 can be most likely attributed to pre-processing differences between the original tensorflow implementation Xie et al. (2020a) and the PyTorch reimplementation we employ. We start with slightly weaker baselines for ResNet50 and ResNext101 than Schneider et al. (2020): ResNet50 and ResNext101 results are slightly worse than previously reported results (typically 0.1% points) due to the smaller batch size of 128 and 32. Smaller batch sizes impact the quality of re-estimated batch norm statistics when computation is performed on the fly Schneider et al. (2020), which is of no concern here due to the large gains obtained by pseudo-labeling.

| | gauss | shot | impulse | defocus | glass | motion | zoom | snow | frost | fog | bright | contrast | elastic | pixelate | jpeg | **mCE** |
|---|---|---|---|---|---|---|---|---|---|---|---|---|---|---|---|---|
| ResNet50 | | | | | | | | | | | | | | | | |
| Baseline (Schneider et al., 2020) | | | | | | | | | | | | | | | | 62.2 |
| Baseline (ours) | 57.2 | 59.5 | 60.0 | 61.4 | 62.3 | 51.3 | 49.5 | 54.6 | 54.1 | 39.3 | 29.1 | 46.7 | 41.4 | 38.2 | 41.8 | 62.8 |
| ENT | 45.5 | 45.5 | 46.8 | 48.4 | 48.7 | 40.0 | 40.3 | 42.0 | 46.6 | 33.2 | 28.1 | 42.4 | 35.2 | 32.2 | 35.1 | 51.6 |
| RPL | 44.2 | 44.4 | 45.5 | 47.0 | 47.4 | 38.8 | 39.2 | 40.7 | 46.2 | 32.5 | 27.7 | 42.7 | 34.6 | 31.6 | 34.4 | 50.5 |
| ResNeXt101 Baseline | | | | | | | | | | | | | | | | |
| Baseline (Schneider et al., 2020) | | | | | | | | | | | | | | | | 56.7 |
| Baseline (ours) | 52.8 | 54.1 | 54.0 | 55.4 | 56.8 | 46.7 | 46.6 | 48.5 | 49.4 | 36.6 | 25.4 | 42.8 | 37.8 | 32.5 | 36.7 | 56.8 |
| ENT | 40.5 | 39.5 | 41.4 | 41.6 | 43.0 | 34.1 | 34.5 | 35.0 | 39.4 | 28.5 | 24.0 | 33.8 | 30.3 | 27.2 | 30.5 | 44.3 |
| RPL | 39.4 | 38.9 | 39.8 | 40.3 | 41.0 | 33.4 | 33.8 | 34.6 | 38.7 | 28.0 | 23.7 | 31.4 | 29.8 | 26.8 | 30.0 | 43.2 |
| ResNeXt101 IG-3.5B | | | | | | | | | | | | | | | | |
| Baseline (Schneider et al., 2020) | | | | | | | | | | | | | | | | 51.6 |
| Baseline (ours) | 50.7 | 51.5 | 53.1 | 54.2 | 55.5 | 45.5 | 44.7 | 41.7 | 42.0 | 28.1 | 20.1 | 33.8 | 35.4 | 27.8 | 33.9 | 51.8 |
| ENT | 38.6 | 38.3 | 40.4 | 41.4 | 41.5 | 33.8 | 33.6 | 32.2 | 34.6 | 24.1 | 19.7 | 26.3 | 27.6 | 24.2 | 27.9 | 40.8 |
| RPL | 39.1 | 39.2 | 40.8 | 42.1 | 42.4 | 33.7 | 33.5 | 31.8 | 34.7 | 23.9 | 19.6 | 26.1 | 27.5 | 23.8 | 27.5 | 40.9 |
| ResNeXt101 DeepAug+Augmix | | | | | | | | | | | | | | | | |
| Baseline (Schneider et al., 2020) | | | | | | | | | | | | | | | | 38.0 |
| Baseline (ours) | 30.0 | 30.0 | 30.2 | 32.9 | 35.5 | 28.9 | 31.9 | 33.3 | 32.8 | 29.5 | 22.6 | 28.4 | 31.2 | 23.0 | 26.5 | 38.1 |
| ENT | 28.7 | 28.5 | 29.0 | 29.8 | 30.9 | 26.9 | 28.0 | 29.3 | 30.5 | 26.2 | 23.2 | 26.3 | 28.5 | 23.7 | 26.0 | 35.5 |
| RPL | 28.1 | 27.8 | 28.3 | 29.1 | 30.1 | 26.3 | 27.4 | 28.8 | 29.8 | 25.9 | 22.7 | 25.6 | 27.9 | 23.2 | 25.4 | 34.8 |
| Noisy Student L2 | | | | | | | | | | | | | | | | |
| Baseline (Xie et al., 2020a) | | | | | | | | | | | | | | | | 28.3 |
| Baseline (ours) | 21.6 | 22.0 | 20.5 | 23.9 | 40.5 | 19.8 | 23.2 | 22.8 | 26.9 | 21.0 | 15.2 | 21.2 | 24.8 | 17.9 | 18.6 | 28.9 |
| ENT | 18.5 | 18.7 | 17.4 | 18.8 | 23.4 | 16.9 | 18.8 | 17.1 | 19.6 | 16.8 | 14.1 | 16.6 | 19.6 | 15.8 | 16.5 | 23.0 |
| RPL | 17.8 | 18.0 | 17.0 | 18.1 | 21.4 | 16.4 | 17.9 | 16.4 | 18.7 | 15.7 | 13.6 | 15.6 | 19.2 | 15.0 | 15.6 | 22.0 |

## C.4 DETAILED RESULTS FOR THE CIFAR10-C AND UDA ADAPTATION

Table 19: Detailed results for each corruption along with mean error on CIFAR10-C as reported in Table 2 in the main paper.

| | gauss | shot | impulse | defocus | glass | motion | zoom | snow | frost | fog | bright | contrast | elastic | pixelate | jpeg | **avg** |
|---|---|---|---|---|---|---|---|---|---|---|---|---|---|---|---|---|
| WRN-28-10 vanilla | | | | | | | | | | | | | | | | |
| Baseline | 53.0 | 41.2 | 44.7 | 18.5 | 49.0 | 22.3 | 24.4 | 18.1 | 25.0 | 11.2 | 6.7 | 17.4 | 16.2 | 28.0 | 22.4 | 26.5 |
| BN adapt | 20.8 | 17.6 | 22.7 | 8.1 | 28.4 | 10.9 | 9.2 | 14.2 | 13.0 | 8.7 | 6.8 | 8.5 | 13.5 | 12.1 | 21.0 | 14.4 |
| ENT | 18.5 | 15.9 | 20.6 | 7.8 | 25.5 | 10.6 | 8.5 | 13.1 | 12.3 | 8.3 | 6.9 | 8.0 | 12.6 | 11.1 | 18.9 | 13.3 |
| RPL | 19.6 | 16.7 | 21.9 | 8.1 | 27.1 | 10.9 | 8.9 | 13.9 | 13.0 | 8.7 | 6.9 | 8.4 | 13.2 | 11.7 | 20.1 | 13.9 |
| WRN-40-2 AM | | | | | | | | | | | | | | | | |
| Baseline | 19.1 | 14.0 | 13.3 | 6.3 | 17.1 | 7.9 | 7.0 | 10.4 | 10.6 | 8.5 | 5.9 | 9.7 | 9.2 | 16.8 | 11.9 | 11.2 |
| BN adapt | 14.1 | 11.9 | 13.9 | 7.2 | 17.6 | 8.7 | 7.9 | 10.8 | 10.6 | 9.0 | 6.8 | 9.0 | 10.9 | 10.1 | 14.0 | 10.8 |
| TENT | 10.8 | 9.1 | 10.9 | 6.0 | 13.4 | 7.2 | 6.3 | 8.4 | 7.8 | 7.1 | 5.7 | 7.1 | 9.2 | 7.4 | 11.2 | 8.5 |
| RPL | 12.4 | 10.5 | 12.4 | 6.5 | 15.6 | 7.8 | 6.9 | 9.5 | 9.1 | 8.2 | 6.2 | 8.3 | 9.9 | 8.8 | 12.8 | 9.7 |
| WRN-26-16 UDA-SS | | | | | | | | | | | | | | | | |
| Baseline | 26.0 | 24.7 | 19.3 | 22.4 | 56.2 | 32.4 | 32.1 | 31.7 | 31.2 | 26.6 | 15.8 | 20.4 | 26.3 | 21.5 | 28.9 | 27.7 |
| BN adapt | 20.5 | 19.0 | 15.6 | 13.5 | 43.1 | 19.4 | 18.3 | 23.1 | 21.2 | 16.2 | 12.8 | 14.1 | 20.9 | 16.7 | 23.4 | 19.9 |
| ENT | 16.9 | 16.7 | 12.3 | 11.3 | 37.6 | 15.6 | 14.8 | 18.3 | 18.2 | 13.4 | 10.8 | 11.9 | 17.9 | 14.4 | 20.9 | 16.7 |
| RPL | 18.1 | 17.1 | 13.2 | 11.9 | 41.5 | 17.3 | 16.1 | 20.4 | 19.1 | 14.5 | 11.8 | 12.7 | 18.8 | 18.1 | 22.6 | 18.2 |

Table 20: Detailed results for the UDA methods reported in Table 2 of the main paper.

| | Baseline | BN adapt | RPL | ENT |
|---|---|---|---|---|
| UDA CIFAR10→STL10, top1 error on target [%]($\searrow$) | | | | |
| WRN-26-16 UDA-SS | 28.7 | 24.6 | 22.9 | 21.8 |
| WRN-26-16 DANN | 25.0 | 25.0 | 24.0 | 23.9 |
| UDA MNIST→MNIST-M, top1 error on target [%]($\searrow$) | | | | |
| WRN-26-16 UDA-SS | 4.8 | 3.9 | 2.4 | 2.0 |
| WRN-26-2 DANN | 11.4 | 6.2 | 5.2 | 5.1 |

### C.5 ABLATION OVER THE HYPERPARAMETER $q$ FOR RPL

For RPL, we must choose the hyperparameter $q$. We performed an ablation study over $q$ and show results in Table 21, demonstrating that RPL is robust to the choice of $q$, with slight preference to higher values. Note: In the initial parameter sweep for this paper, we only compared $q = 0.7$ and $q = 0.8$. Given the result in Table 21, it could be interesting to re-run the models in Table 1 of the main paper with $q = 0.9$, which could yield another (small) improvement in mCE.

Table 21: ImageNet-C dev set mCE in %, vanilla ResNet50, batch size 96. We report the best score across a maximum of six adaptation epochs.

| q | 0.5 | 0.6 | 0.7 | 0.8 | 0.9 |
|---|---|---|---|---|---|
| mCE (dev) | 49.5 | 49.3 | 49.2 | 49.2 | 49.1 |

### C.6 SELF-TRAINING OUTPERFORMS CONTRASTIVE TEST-TIME TRAINING (SUN ET AL., 2019B)

Sun et al. (2019b) use a ResNet18 for their experiments on ImageNet and only evaluate their method on severity 5 of IN-C. To enable a fair comparison, we trained a ResNet18 with both hard labeling and RPL and compare the efficacy of both methods to Test-Time Training in Table 22. For both hard labeling and RPL, we use the hyperparameters we found for the vanilla ResNet50 model and thus, we expect even better results for hyperparameters tuned on the vanilla ResNet18 model and following our general hyperparameter search protocol.

While all methods (self-learning and TTT) improve the performance over a simple vanilla ResNet18, we note that even the very simple baseline using hard labeling already outperfoms Test-Time Training; further gains are possible with RPL. The result highlights the importance of simple baselines (like self-learning) when proposing new domain adaptation schemes. It is likely that many established DA techniques more complex than the basic self-learning techniques considered in this work will even further improve over TTT and other adaptation approaches developed exclusively in robustness settings.

Table 22: Comparison of hard-pseudo labeling and robust pseudo-labeling to Test-Time Training Sun et al. (2019b): Top-1 error for a ResNet18 and severity 5 for all corruptions. Simple hard pseudo-labeling already outperforms TTT, robust pseudo labeling over multiple epochs yields additional gains.

| | gauss | shot | impulse | defocus | glass | motion | zoom | snow | frost | fog | bright | contrast | elastic | pixelate | jpeg | **Avg** |
|---|---|---|---|---|---|---|---|---|---|---|---|---|---|---|---|---|
| vanilla ResNet18 | 98.8 | 98.2 | 99.0 | 88.6 | 91.3 | 88.8 | 82.4 | 89.1 | 83.5 | 85.7 | 48.7 | 96.6 | 83.2 | 76.9 | 70.4 | 85.4 |
| Test-Time Training | 73.7 | 71.4 | 73.1 | 76.3 | 93.4 | 71.3 | 66.6 | 64.4 | 81.3 | 52.4 | 41.7 | **64.7** | 55.7 | 52.2 | 55.7 | 66.3 |
| hard PL, (1 epoch) | 73.2 | 70.8 | 73.6 | 76.5 | 75.6 | 63.9 | 56.1 | 59.0 | **65.9** | 48.4 | 39.7 | 85.2 | 50.4 | 47.0 | 51.5 | 62.5 |
| RPL (4 epochs) | **71.3** | **68.3** | **71.7** | **76.2** | **75.6** | **61.5** | **54.4** | **56.9** | 67.1 | **47.3** | **39.3** | 93.2 | **48.9** | **45.7** | **50.4** | **61.9** |

## C.7 EFFECT OF BATCH SIZE AND LINEAR LEARNING RATE SCALING

How is self-learning performance affected by batch size constraints? We compare the effect of different batch sizes and linear learning rate scaling. In general, we found that affine adaptation experiments on ResNet50 scale can be run with batch size 128 on a Nvidia V100 GPU (16GB), while only batch size 96 experiments are possible on RTX 2080 GPUs.

The results in Table 23 show that for a ResNet50 model, higher batch size yields a generally better performance.

Table 23: ImageNet-C dev set mCE for various batch sizes with linear learning rate scaling. All results are computed for a vanilla ResNet50 model using RPL with $q = 0.8$, reporting the best score across a maximum of six adaptation epochs.

| batch size | 16 | 32 | 64 | 80 | 96 | 128 |
|---|---|---|---|---|---|---|
| learning rate ($\times 10^{-3}$) | 0.125 | 0.250 | 0.500 | 0.625 | 0.750 | 1 |
| dev mCE | 53.8 | 51.0 | 49.7 | 49.3 | 49.2 | 48.9 |

## C.8 PERFORMANCE OVER DIFFERENT SEEDS IN A RESNET50 ON IMAGENET-C

To limit the amount of compute, we ran RPL and ENT for our vanilla ResNet50 model three times with the optimal hyperparameters. The averaged results, displayed as "mean (unbiased std)" are:

Table 24: ImageNet-C performance for three seeds on a ResNet50 for ENT and RPL.

| ResNet50 + self-learning | mCE on IN-C dev [%] | mCE on IN-C test [%] |
|---|---|---|
| ENT | 50.0 (0.04) | 51.6 (0.04) |
| RPL | 48.9 (0.02) | 50.5 (0.03) |

## C.9 SELF-LEARNING AS CONTINUOUS TEST-TEST ADAPTATION

We test our method on continuous test-time adaptation where the model adapts to a continuous stream of data from the same domain. In Fig. 7, we display the error of the Noisy Student L2 model while it is being adapted to ImageNet-C and ImageNet-R. The model performance improves as the model sees more data from the new domain. We differentiate continuous test-time adaptation from the online test-time adaptation setting (Zhang et al., 2021) where the model is adapted to each test sample individually, and reset after each test sample.

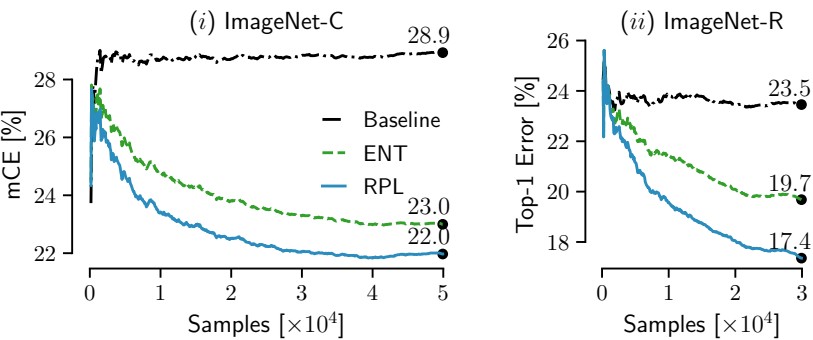

Figure 7: Evolution of error during online adaptation for EfficientNet-L2.

# D   DETAILED AND ADDITIONAL RESULTS ON IN-D

## D.1   EVALUATION PROTOCOL ON IN-D

The domains in IN-D differ in terms of their difficulty for the studied models. Therefore, to calculate an aggregate score, we propose normalizing the error rates by the error achieved by AlexNet on the respective domains to calculate the mean error, following the approach in Hendrycks & Dietterich (2019) for IN-C. This way, we obtain the aggregate score mean Domain Error (mDE) by calculating the mean over different domains,

$$\text{DE}_d^f = \frac{E_d^f}{E_d^{\text{AlexNet}}}, \qquad \text{mDE} = \frac{1}{D} \sum_{d=1}^{D} E_d^f, \tag{21}$$

where $E_d^f$ is the top-1 error of a classifier $f$ on domain $d$.

**Leave-one-out-cross-validation**   For all IN-D results we report in this paper, we chose the hyperparameters on the IN-C dev set. We tried a different model selection scheme on IN-D as a control experiment with "Leave one out cross-validation" (L1outCV): with a round-robin procedure, we choose the hyperparameters for the test domain on all other domains. We select the same hyperparameters as when tuning on the "dev" set: For the ResNet50 model, we select over the number of training epochs (with a maximum of 7 training epochs) and search for the optimal learning rate in the set [0.01, 0.001, 0.0001]. For the EfficientNet-L2 model, we train only for one epoch as before and select the optimal learning rate in the set [$4.6 \times 10^{-3}$, $4.6 \times 10^{-4}$, $4.6 \times 10^{-5}$, $4.6 \times 10^{-6}$]. This model selection leads to worse results both for the ResNet50 and the EfficientNet-L2 models, highlighting the robustness of our model selection process, see Table 25.

Table 25: mDE in % on IN-D for different model selection strategies.

| model | model selection | |
| --- | --- | --- |
| | L1outCV | IN-C dev |
| ResNet50 RPL$_{q=0.8}$ | 81.3 | 76.1 |
| ResNet50 ENT | 82.4 | 77.3 |
| EfficientNet-L2 ENT | 69.2 | 66.8 |
| EfficientNet-L2 RPL$_{q=0.8}$ | 69.1 | 67.2 |

## D.2   DETAILED RESULTS FOR ROBUST RESNET50 MODELS ON IN-D

We show detailed results for all models on IN-D for vanilla evaluation (Table 26) BN adaptation (Table 27), RPL$_{q=0.8}$ (Table 28) and ENT(Table 29). For RPL$_{q=0.8}$ and ENT, we use the same hyperparameters that we chose on our IN-C 'dev' set. This means we train the models for 5 epochs with RPL$_{q=0.8}$ and for one epoch with ENT.

We evaluate the pre-trained and public checkpoints of SIN (Geirhos et al., 2019), ANT (Rusak et al., 2020), ANT+SIN (Rusak et al., 2020), AugMix (Hendrycks et al., 2020b), DeepAugment (Hendrycks et al., 2020a) and DeepAug+Augmix (Hendrycks et al., 2020a) in the following tables.

Table 26: Top-1 error on IN-D in % as obtained by robust ResNet50 models. For reference, we also show the mCE on IN-C and the top-1 error on IN-R. See main test for model references.

| Model | Clipart | Infograph | Painting | Quickdraw | Real | Sketch | mDE | IN-C | IN-R |
| --- | --- | --- | --- | --- | --- | --- | --- | --- | --- |
| vanilla | 76.0 | 89.6 | 65.1 | 99.2 | 40.1 | 82.0 | 88.2 | 76.7 | 63.9 |
| SIN | 71.3 | 88.6 | 62.6 | 97.5 | 40.6 | 77.0 | 85.6 | 69.3 | 58.5 |
| ANT | 73.4 | 88.9 | 63.3 | 99.2 | 39.9 | 80.8 | 86.9 | 62.4 | 61.0 |
| ANT+SIN | 68.4 | 88.6 | 60.6 | 95.5 | 40.8 | 70.3 | 83.1 | 60.7 | 53.7 |
| AugMix | 70.8 | 88.6 | 62.1 | 99.1 | 39.0 | 78.5 | 85.4 | 65.3 | 58.9 |
| DeepAugment | 72.0 | 88.8 | 61.4 | 98.9 | 39.4 | 78.5 | 85.6 | 60.4 | 57.8 |
| DeepAug+Augmix | 68.4 | 88.1 | 58.7 | 98.2 | 39.2 | 75.2 | 83.4 | 53.6 | 53.2 |

Table 29: Top-1 error on IN-D in % as obtained by state-of-the-art robust ResNet50 models and ENT. See main text for references to the used models.

| Model | Clipart | Infograph | Painting | Quickdraw | Real | Sketch | mDE |
|---|---|---|---|---|---|---|---|
| vanilla | 65.1 | 85.8 | 59.2 | 98.5 | 38.4 | 75.8 | 77.3 |
| SIN | 62.1 | 87.0 | 57.3 | 99.1 | 39.0 | 68.6 | 75.5 |
| ANT | 64.2 | 86.9 | 58.7 | 97.1 | 38.8 | 72.8 | 76.5 |
| ANT+SIN | 62.2 | 86.8 | 57.7 | 95.8 | 40.1 | 68.7 | 75.2 |
| AugMix | 60.2 | 84.6 | 55.8 | 97.6 | 36.8 | 72.0 | 74.4 |
| DeepAugment | 59.5 | 85.7 | 54.4 | 98.0 | 37.1 | 66.4 | 73.3 |
| DeepAug+Augmix | 58.4 | 84.3 | 54.7 | 98.5 | 38.1 | 63.6 | 72.7 |

Table 30: mDE on IN-D in % as obtained by robust ResNet50 models with a baseline evaluation, batch norm adaptation, $RPL_{q=0.8}$ and ENT. See main text for model references.

| | mDE on IN-D ($\searrow$) | | | |
|---|---|---|---|---|
| Model | Baseline | BN adapt | $RPL_{q=0.8}$ | ENT |
| vanilla | 88.2 | 80.2 | 76.1 | 77.3 |
| SIN | 85.6 | 79.6 | 76.8 | 75.5 |
| ANT | 86.9 | 80.7 | 78.1 | 76.5 |
| ANT+SIN | **83.1** | 77.8 | 76.1 | 75.2 |
| AugMix | 85.4 | 78.4 | 74.6 | 74.4 |
| DeepAugment | 85.6 | 78.8 | 74.8 | 73.3 |
| DeepAugment+Augmix | 83.4 | **74.9** | **72.6** | **72.7** |

Table 27: Top1 error on IN-D in % as obtained by state-of-the-art robust ResNet50 models and batch norm adaptation, with a batch size of 128. See main text for model references.

| Model | Clipart | Infograph | Painting | Quickdraw | Real | Sketch | mDE |
|---|---|---|---|---|---|---|---|
| vanilla | 70.2 | 88.2 | 63.5 | 97.8 | 41.1 | 78.3 | 80.2 |
| SIN | 67.3 | 89.7 | 62.2 | 97.2 | 44.0 | 75.2 | 79.6 |
| ANT | 69.2 | 89.4 | 63.0 | 97.5 | 42.9 | 79.5 | 80.7 |
| ANT+SIN | 64.9 | 88.2 | 60.0 | 96.8 | 42.6 | 73.0 | 77.8 |
| AugMix | 66.9 | 88.1 | 61.2 | 97.1 | 40.4 | 75.0 | 78.4 |
| DeepAugment | 66.6 | 89.7 | 60.0 | 97.2 | 42.5 | 75.1 | 78.8 |
| DeepAug+Augmix | 61.9 | 85.7 | 57.5 | 95.3 | 40.2 | 69.2 | 74.9 |

Table 28: Top-1 error on IN-D in % as obtained by state-of-the-art robust ResNet50 models and $RPL_{q=0.8}$. See main text for model references.

| Model | Clipart | Infograph | Painting | Quickdraw | Real | Sketch | mDE |
|---|---|---|---|---|---|---|---|
| vanilla | 63.6 | 85.1 | 57.8 | 99.8 | 37.3 | 73.0 | 76.1 |
| SIN | 60.8 | 86.4 | 56.0 | 99.0 | 37.8 | 67.0 | 76.8 |
| ANT | 63.4 | 86.3 | 57.7 | 99.2 | 37.7 | 71.0 | 78.1 |
| ANT+SIN | 61.5 | 86.4 | 56.8 | 97.0 | 39.0 | 67.1 | 76.1 |
| AugMix | 59.7 | 83.4 | 54.1 | 98.2 | 35.6 | 70.1 | 74.6 |
| DeepAugment | 58.1 | 84.6 | 53.3 | 99.0 | 36.2 | 64.2 | 74.8 |
| DeepAug+Augmix | 57.0 | 83.2 | 53.4 | 99.1 | 36.5 | 61.3 | 72.6 |

The summary results for all models are shown in Table 30.

We show the top-1 error for the different IN-D domains versus training epochs for a vanilla ResNet50 in Fig. 8. We indicate the epochs 1 and 5 at which we extract the errors with dashed black lines.

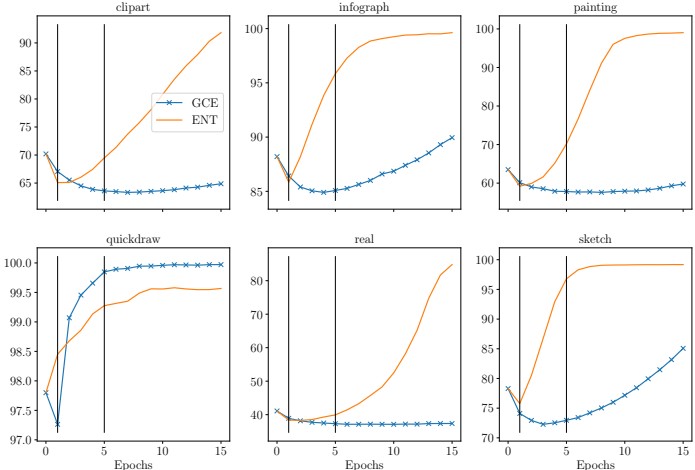

Figure 8: Top-1 error for the different IN-D domains for a ResNet50 and training with RPL$_{q=0.8}$ and ENT. We indicate the epochs at which we extract the test errors by the dashed black lines (epoch 1 for ENTand epoch 5 for RPL$_{q=0.8}$).

### D.3 DETAILED RESULTS FOR THE EFFICIENTNET-L2 NOISY STUDENT MODEL ON IN-D

We show the detailed results for the EfficientNet-L2 Noisy Student model on IN-D in Table 31.

Table 31: Top-1 error ($\searrow$) on IN-D in % for EfficientNet-L2

| Domain | Baseline | ENT | RPL |
|---|---|---|---|
| Clipart | 45.0 | 39.8 | **37.9** |
| Infograph | **77.9** | 91.3 | 94.3 |
| Painting | 42.7 | 41.7 | **40.9** |
| Quickdraw | **98.4** | 99.4 | 99.4 |
| Real | 29.2 | 28.7 | **27.9** |
| Sketch | 56.4 | **48.0** | 51.5 |
| mDE | 67.2 | **66.8** | 67.2 |

### D.4 DETAILED RESULTS ON THE ERROR ANALYSIS ON IN-D

**Frequently predicted classes** We analyze the most frequently predicted classes on IN-D by a vanilla ResNet50 and show the results in Fig. 9. We make several interesting observations: First, we find most errors interpretable: it makes sense that a ResNet50 assigns the label "comic book" to images from the "clipart" or "painting" domains, or "website" to images from the "infograph" domain, or "envelope" to images from the "sketch" domain. Second, on the hard domain "quickdraw", the ResNet50 mostly predicts non-sensical classes that are not in IN-D, mirroring its almost chance performance on this domain. Third, we find no systematic errors on the "real" domain which is expected since this domain should be similar to IN.

**Filtering predictions on IN-D that cannot be mapped to ImageNet-D** We perform a second analysis: We filter the predicted labels according to whether they can be mapped to IN-D and report the filtered top-1 errors as well as the percentage of filtered out inputs in Table 32. We note that for the domains "infograph" and "quickdraw", the ResNet50 predicts labels that cannot be mapped to IN-D in over 70% of all cases, highlighting the hardness of these two domains.

Table 32: top-1 error on IN and different IN-D domains for different settings: left column: default evaluation, middle column: predicted labels that cannot be mapped to IN-D are filtered out, right column: percentage of filtered out labels.

| Dataset | top-1 error in % | top-1 error on filtered labels in % | percentage of rejected inputs |
|---|---|---|---|
| IN val | 12.1 | 13.4 | 52.7 |
| IN-D real | 40.2 | 17.2 | 27.6 |
| IN-D clipart | 76.1 | 59.0 | 59.0 |
| IN-D infograph | 89.7 | 59.3 | 74.6 |
| IN-D painting | 65.2 | 39.5 | 42.4 |
| IN-D quickdraw | 99.3 | 96.7 | 76.1 |
| IN-D sketch | 82.1 | 65.6 | 47.9 |

**Filtering labels and predictions on IN that cannot be mapped to ImageNet-D**  To test for possible class-bias effects, we test the performance of a ResNet50 model on IN classes that can be mapped to IN-D and report the results in Table 32.

First, we map IN labels to IN-D to make the setting as similar as possible to our experiments on IN-D and report the top-1 error (12.1%). This error is significantly lower compared to the top-1 error a ResNet50 obtains following the standard evaluation protocol (23.9%). This can be explained by the simplification of the task: While in IN there are 39 bird classes, these are all mapped to the same hierarchical class in IN-D. Therefore, the classes in IN-D are more dissimilar from each other than in IN. Additionally, there are only 164 IN-D classes compared to the 1000 IN classes, raising the chance level prediction.

If we further only accept predictions that can be mapped to IN-D, the top-1 error is slightly increased to 13.4%. In total, about 52.7% of all images in the IN validation set cannot be mapped to IN-D.

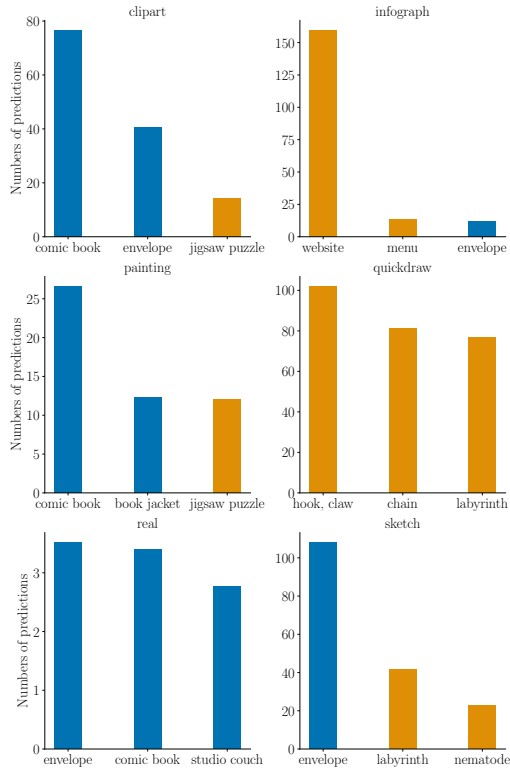

Figure 9: Systematic predictions of a vanilla ResNet50 on IN-D for different domains.

## D.5 Top-1 error on IN-D for AlexNet

We report the top-1 error numbers on different IN-D as achieved by AlexNet in Table 33. We used these numbers for normalization when calculating mDE.

Table 33: top-1 error on IN-D by AlexNet which was used for normalization.

| Dataset | top-1 error in % |
|---|---|
| IN-D real | 54.887 |
| IN-D clipart | 84.010 |
| IN-D infograph | 95.072 |
| IN-D painting | 79.080 |
| IN-D quickdraw | 99.745 |
| IN-D sketch | 91.189 |

# E  ADDITIONAL EXPERIMENTS

## E.1  BEYOND IMAGENET CLASSES: SELF-LEARNING ON WILDS

The WILDS benchmark (Koh et al., 2021) is comprised of ten tasks to test domain generalization, subpopulation shift, and combinations thereof. In contrast to the setting considered here, many of the datasets in WILDS mix several 10s or 100s domains during test time.

The Camelyon17 dataset in WILDS contains histopathological images, with the labels being binary indicators of whether the central $32\times32$ region contains any tumor tissue; the domain identifies the hospital that the patch was taken from. Camelyon17 contains three different test splits with different domains and varying difficulty levels. For evaluation, we took the pretrained checkpoint from worksheets.codalab.org/worksheets/0x00d14c55993548a1823a710642f6d608 (camelyon17_erm_densenet121_seed0) for a DenseNet121 model (Huang et al., 2017) and verified the reported baseline performance numbers. We adapt the models using ENT or RPL for a maximum of 10 epochs using learning rates $\{3\times10^{-5}, 3\times10^{-4}, \ldots 3\times10^{-1}\}$. The best hyperparameter is selected according to OOD Validation accuracy.

The RxRx1 dataset in WILDS contains RGB images of cells obtained by fluorescent microscopy, with the labels indicating which of the 1,139 genetic treatments (including no treatment) the cells received; the domain identifies the batch in which the imaging experiment was run. The RxRx1 dataset contains three test splits, however, unlike Camelyon17, in all of the splits the domains are mixed. For evaluation, we took the pretrained checkpoint from worksheets.codalab.org/bundles/0x7d33860545b64acca5047396d42c0ea0 for a ResNet50 model and verified the reported baseline performance numbers. We adapt the models using ENT or RPL for a maximum of 10 epochs using base learning rates $\{6.25\times10^{-6}, 6.25\times10^{-5}, \ldots 6.25\times10^{-2}\}$, which are scaled to the admissible batch size for single GPU adaptation using linear scaling. The best hyperparameter is selected according to OOD Validation accuracy.

Table 34: Self-learning can improve performance on WILDS if a systematic shift is present — on Camelyon17, the ood validation and test sets are different hospitals, for example. On datasets like RxRx1 and FMoW, we do not see an improvement, most likely because the ood domains are shuffled, and a limited amount of images exist for each test domain.

|  | Validation (ID) | Top-1 accuracy [%] Validation (OOD) | Test (OOD) |
|---|---|---|---|
| Camelyon17 |  |  |  |
| Baseline | 81.4 | 88.7 | 63.1 |
| BN adapt | 97.8 (+16.4) | 90.9 (+2.2) | 88.0 (+24.9) |
| ENT | 97.6 (+16.2) | 92.7 (+4.0) | 91.6 (+28.5) |
| RPL | 97.6 (+16.2) | 93.0 (+4.3) | 91.0 (+27.9) |
| RxRx1 |  |  |  |
| Baseline | 35.9 | 19.1 | 29.7 |
| BN adapt | 35.0 (-0.9) | 19.1 (0.0) | 29.4 (-0.3) |
| ENT | 34.8 (-1.1) | 19.2 (+0.1) | 29.4 (-0.3) |
| RPL | 34.8 (-1.1) | 19.2 (+0.1) | 29.4 (-0.3) |
| FMoW |  |  |  |
| Baseline | 60.5 | 59.2 | 52.9 |
| BN adapt | 59.9 (-0.6) | 57.6 (-1.6) | 51.8 (-1.1) |
| ENT | 59.9 (-0.6) | 58.5 (-0.7) | 52.2 (-0.7) |
| RPL | 59.8 (-0.7) | 58.6 (-0.6) | 52.1 (-0.8) |

The FMoW dataset in WILDS contains RGB satellite images, with the labels being one of 62 building or land use categories; the domain specifies the year in which the image was taken and its geographical region (Africa, the Americas, Oceania, Asia, or Europe). The FMoW dataset contains four test splits for different time periods, for which all regions are mixed together. For evaluation, we took the pretrained checkpoint from //worksheets.codalab.org/bundles/0x20182ee424504e4a916fe88c91afd5a2 for a DenseNet121 model and verified the reported baseline performance numbers. We adapt the models using ENT or RPL for a maximum of 10 epochs

using learning rates $\{5.0 \times 10^{-6}, 5.0 \times 10^{-5}, \ldots 5.0 \times 10^{-2}\}$. The best hyperparameter is selected according to OOD Validation accuracy.

While we see improvements on Camelyon17, neither BN adaptation nor self-learning can improve performance on RxRx1 or FMoW. Initial experiments on PovertyMap and iWildsCam also do not show improvements with self-learning. We hypothesize that the reason lies in the mixing of the domains: Both BN adaptation and our self-learning methods work best on systematic domain shifts. These results support our claim that self-learning is effective, while showing the important limitation when applied to more diverse shifts.

### E.2    SMALL IMPROVEMENTS ON BIGTRANSFER MODELS WITH GROUP NORMALIZATION LAYERS

We evaluated BigTransfer models (Kolesnikov et al., 2020) provided by the timm library (Wightman, 2019). A difference to the ResNet50, ResNeXt101 and EfficientNet models is the use of group normalization layers, which might influence the optimal method for adaptation—for this evaluation, we followed our typical protocol as performed on ResNet50 models, and used affine adaptation.

For affine adaptation, a distilled BigTransfer ResNet50 model improves from 49.6 % to 48.4 % mCE on the ImageNet-C development set, and from 55.0 % to 54.4 % mCE on the ImageNet-C test set when using RPL ($q = 0.8$) for adaptation, at learning rate $7.5 \times 10^{-4}$ at batch size 96 after a single adaptation epoch. Entropy minimization did not further improve results on the ImageNet-C test set. An ablation over learning rates and epochs on the dev set is shown in Table 35, the final results are summarized in Table 36.

Table 35: mCE in % on the IN-C dev set for ENT and RPL for different numbers of training epochs when adapting the affine batch norm parameters of a ResNet50 model.

| criterion | ENT | | | RPL | | |
|---|---|---|---|---|---|---|
| lr, 7.5 × epoch | $10^{-5}$ | $10^{-4}$ | $10^{-3}$ | $10^{-5}$ | $10^{-4}$ | $10^{-3}$ |
| 0 | 49.63 | 49.63 | 49.63 | 49.63 | 49.63 | 49.63 |
| 1 | 49.44 | 50.42 | 52.59 | 49.54 | 48.89 | 48.95 |
| 2 | 49.26 | 50.27 | 56.47 | 49.47 | **48.35** | 50.77 |
| 3 | 49.08 | 52.18 | 60.06 | 49.39 | 48.93 | 51.45 |
| 4 | 48.91 | 52.03 | 60.50 | 49.31 | 50.01 | 51.53 |
| 5 | **48.80** | 51.97 | 62.91 | 49.24 | 49.96 | 51.34 |
| 6 | 48.83 | 52.10 | 62.96 | 49.16 | 49.71 | 51.19 |
| 7 | 48.83 | 52.10 | 62.96 | 49.16 | 49.71 | 51.19 |

Table 36: mCE in % on the IN-C dev set for ENT and RPL for different numbers of training epochs when adapting the affine batch norm parameters of a ResNet50 model.

| | dev mCE | test mCE |
|---|---|---|
| Baseline | 49.63 | 55.03 |
| ENT | 48.80 | 56.36 |
| RPL | **48.35** | **54.41** |

### E.3    CAN SELF-LEARNING IMPROVE OVER SELF-LEARNING BASED UDA?

An interesting question is whether test-time adaptation with self-learning can improve upon self-learning based UDA methods. To investigate this question, we build upon French et al. (2018) and their released code base at github.com/Britefury/self-ensemble-visual-domain-adapt. We trained the Baseline models from scratch using the provided shell scripts with the default hyperparameters and verified the reported performance. For adaptation, we tested BN adaptation, ENT, RPL, as well as continuing to train in exactly the setup of French et al. (2018), but without the supervised loss. For the different losses, we adapt the models for a maximum of 10 epochs using learning rates $\{1 \times 10^{-5}, 1 \times 10^{-4}, \ldots, 1 \times 10^{-1}\}$.

Note that for this experiment, in contrast to any other result in this paper, **we purposefully do not perform proper hyperparameter selection based on a validation dataset**—instead we report the best accuracy across all tested epochs and learning rates to give an upper bound on the achievable performance for test-time adaptation.

As highlighted in Table 37, none of the four tested variants is able to meaningfully improve over the baseline, corroborating our initial hypothesis that self-learning within a full UDA setting is the optimal strategy, if dataset size and compute permits. On the other hand, results like the teacher

refinement step in DIRT-T (Shu et al., 2018) show that with additional modifications in the loss function, it might be possible to improve over standard UDA with additional adaptation at test time.

Table 37: Test-time adaptation marginally improves over self-ensembling.

| | Baseline | BN adapt | ENT | RPL | Self-ensembling loss |
|---|---|---|---|---|---|
| MNIST→SVHN | | | | | |
| MT+TF | 33.88 | 34.44 | 34.87 | 35.09 | 33.27 |
| MT+CT* | 32.62 | 34.11 | 34.25 | 34.21 | 33.36 |
| MT+CT+TF | 41.59 | 41.93 | 41.95 | 41.95 | 42.70 |
| MT+CT+TFA | 30.55 | 32.53 | 32.54 | 32.55 | 30.84 |
| SVHN-specific aug. | 97.05 | 96.82 | 96.91 | 96.87 | 97.12 |
| MNIST→USPS | | | | | |
| MT+TF | 98.01 | 97.91 | 97.96 | 97.91 | 98.16 |
| MT+CT* | 88.34 | 88.39 | 88.54 | 88.39 | 88.44 |
| MT+CT+TF | 98.36 | 98.41 | 98.41 | 98.41 | 98.50 |
| MT+CT+TFA | 98.45 | 98.45 | 98.45 | 98.45 | 98.61 |
| SVHN→MNIST | | | | | |
| MT+TF | 98.49 | 98.47 | 98.49 | 98.47 | 99.40 |
| MT+CT* | 88.34 | 88.36 | 88.36 | 88.36 | 89.36 |
| MT+CT+TF | 99.51 | 99.49 | 99.5 | 99.49 | 99.57 |
| MT+CT+TFA | 99.56 | 99.57 | 99.57 | 99.57 | 99.58 |
| SVHN-specific aug. | 99.52 | 99.49 | 99.5 | 99.49 | 99.65 |
| USPS→MNIST | | | | | |
| MT+TF | 92.79 | 92.62 | 92.62 | 92.66 | 93.08 |
| MT+CT* | 99.11 | 99.13 | 99.14 | 99.13 | 99.21 |
| MT+CT+TF | 99.41 | 99.42 | 99.45 | 99.42 | 99.52 |
| MT+CT+TFA | 99.48 | 99.54 | 99.57 | 99.54 | 99.54 |

## F    DETAILED DISCUSSION OF RELATED WORK

**Self-learning for domain adaptation**    Xie et al. (2020b) introduce "In-N-Out" which uses auxiliary information to boost both in- and out-of-distribution performance. AdaMatch (Berthelot et al., 2021) builds upon FixMatch (Sohn et al., 2020) and can be used for the tasks of unsupervised domain adaptation, semi-supervised learning and semi-supervised domain adaptation as a general-purpose algorithm. Prabhu et al. (2021) propose SENTRY, an algorithm based on judging the predictive consistency of samples from the target domain under different image transformations. Zou et al. (2019) show that different types of confidence regularization can improve the performance of self-learning. A theoretically motivated framework for self-learning in domain adaptation based on consistency regularization has been proposed by Wei et al. (2020) and then extended by Cai et al. (2021). Self-learning has also been used for semantic segmentation (Zou et al., 2018).

The main difference from these works to ours is that they 1) utilize both source and target data during training (i.e., the classical UDA setup) whereas we only require access to unlabeled target data (source-free setup), and 2) train their models from scratch whereas we adapt pretrained checkpoints to the unlabeled target data, 3) are oftentimes more complicated (also in terms of the number of hyperparameters) than our approach due to using more than one term in the objective function. We would like to highlight that utilizing source data should always result in better performance compared to not using source data. Our contribution is to show that self-learning can still be very beneficial with a small compute budget and no access to source data. Our setup targets "deployed systems", e.g., a self-driving car or a detection algorithm in a production line which adapts to the distribution shift "on-the-fly" and cannot (or should not) be retrained from scratch for every new domain shift.

Kumar et al. (2020) study the setting of self-learning for gradual domain adaptation. They find that self-learning works better if the data distribution changes slowly. The gradual domain adaptation setting differs from ours; instead of a gradual shift over time, we focus on a fixed, systematic shift at test time dataset. Kumar et al. (2020) tested their method on a synthetic Gaussian dataset, MNIST and the Portraits datasets; building and evaluating ImageNet-scale datasets for a gradual domain adaptation perspective is a very interesting extension of our work, but left for future work, and would not only require changes/adaptations to the self-learning method, but also to the evaluation datasets.

Chen et al. (2020b) prove that under certain conditions, self-learning can improve performance in biased datasets where spurious features correlate with the label in the source domain but are independent of the label in the target domain. While Chen et al. (2020b) also consider the setting of source-free domain adaptation (like we do), they limit their experiments to small scale models on MNIST and Celeb-A, while we conduct a large scale empirical study on all common robustness datasets with large scale models – some of the observed and studied effects in our paper (effectiveness of different loss functions at different problem scales, adaptation mechanisms, etc.) can be attributed to this large scale evaluation setting, and extending our insights over small scale experiments.

Similar to us, Chen et al. (2020b) find that a strong source classifier is necessary for self-learning to work; however, in their case, a teacher accuracy of 72% (on CMNIST10) is already too low and leads to worse student accuracy. In contrast, in our experiments, self-learning still works for an mCE as high as 80% (cf. appendix Figure 3, severity 5) and teacher accuracies as low as 10.4% (on ImageNet-D "Infograph"), and breaks down at accuracies around 1-2% (on ImageNet-D "Quickdraw"). This discrepancy might be due to the spurious correlations that Chen et al. (2020b) introduced in their dataset leading to systematic biases, which are not present in the datasets we studied.

**Self-learning in semi-supervised learning (SSL)**    In a different line of work which is not related to domain adaptation directly, self-learning has been used in a semi-supervised setting. Zoph et al. (2020) show that self-learning outperforms pretraining when stronger data augmentation is used and more labeled data is present. They use human labels on the target task (e.g., object detection on COCO) and pseudo-labels on an unlabeled dataset (e.g. ImageNet), and optimize the loss on both datasets, with the aim to improve performance on the task where ground truth labels are known. The work of Zoph et al. (2020) is orthogonal to ours, in the sense that we could adapt their final

checkpoint to a new domain with our method, similar to how we adapted the Noisy Student model which was also trained using self-learning.

Rizve et al. (2021) propose an uncertainty-aware pseudo-label selection (UPS) framework which outperforms other SSL methods in a few-label regime. UPS is helpful to reduce the impact of noisy pseudo-labels; in our case, we use the generalized cross-entropy loss for this purpose. Testing the UPS framework (and other means for improving the quality of pseudo-labels, or robustness against label noise) on robustness datasets would be an interesting direction for future work.

De Sousa Ribeiro et al. (2020) propose Deep Bayesian Self-Training (DBST) for automatic data annotation. Mukherjee & Awadallah (2020) suggest using self-learning in a semi-supervised setting for text classification with few labels.

## G  SOFTWARE STACK

We use different open source software packages for our experiments, most notably Docker (Merkel, 2014), scipy and numpy (Virtanen et al., 2020), GNU parallel (Tange, 2011), Tensorflow (Abadi et al., 2016), PyTorch (Paszke et al., 2017), timm (Wightman, 2019), Self-ensembling for visual domain adaptation (French et al., 2018), the WILDS benchmark (Koh et al., 2021), and torchvision (Marcel & Rodriguez, 2010).

