# OpenReview forum: "If your data distribution shifts, use self-learning"
_ICLR.cc/2022/Conference — ICLR 2022 Submitted_

### Official Review · Reviewer_s85k · 2021-10-31

**Correctness:** 3
**Technical Novelty And Significance:** 3
**Empirical Novelty And Significance:** 4
**Recommendation:** 8
**Confidence:** 3

**Main Review:**

Strengths:

-The message of the paper is very clear, and it is generally well written.

-I believe this paper is of interest to the research community, as improving
the robustness of deep vision models to distributional shifts is of critical importance going forward.

-The empirical evaluation is thorough, significant and convincing, demonstrating
the advantages of using self-training on top of existing approaches to
improve the robustness of deep models on a variety of standard benchmarks.

Weaknesses/Suggestions:

-The insights from this paper are mainly empirical, and there is little
algorithmic/theoretical novelty. The self-training algorithms evaluated here
are well established, even in a deep learning setting (see below).
The exception to this is section 8 which comes across as an attempt to address
this deficiency late on, nonetheless it is useful to know what temperatures
ought to be used in student/teacher training. All this is not to say that
empirical demonstrations are less important, but from an algorithmic perspective
it makes the results not as surprising. At the risk of stating the obvious,
since we're using self-training to adapt to new domains, we would expect the
models to be more robust to them. With that said, demonstrating this
convincingly in practice is non-trivial and demands significant engineering
effort as evidenced in this paper.

-Because there are so many results, the paper is quite dense and a bit hard
to follow at times since the text is broken up a lot. This isn't a major
issue but something the authors could consider improving for the final
version.

-Self-training using deep learning as presented here has gained popularity
recently, and the paper is missing some related references (see below to name a few).

-I think including some results on model calibration would add a lot
of value to the exposition, especially if the authors could demonstrate that
self-learning also calibrates predictions, thereby improving not only accuracy
but also uncertainty under domain shift.

-Reporting standard deviations of results would strengthen the author's claims

-------------------

References:

[1] Zou, Yang, et al. "Unsupervised domain adaptation for semantic segmentation via class-balanced self-training." Proceedings of the European conference on computer vision (ECCV). 2018.

[2] Rizve, Mamshad Nayeem, et al. "In Defense of Pseudo-Labeling: An Uncertainty-Aware Pseudo-label Selection Framework for Semi-Supervised Learning." International Conference on Learning Representations. 2020.

[3] De Sousa Ribeiro, Fabio, et al. "Deep bayesian self-training." Neural Computing and Applications 32.9 (2020): 4275-4291.

[4] Zou, Yang, et al. "Confidence regularized self-training." Proceedings of the IEEE/CVF International Conference on Computer Vision. 2019.

[5] Zoph, Barret, et al. "Rethinking Pre-training and Self-training." Advances in Neural Information Processing Systems 33 (2020).

[6] Mukherjee, Subhabrata, and Ahmed Awadallah. "Uncertainty-aware self-training for few-shot text classification." Advances in Neural Information Processing Systems 33 (2020).

[7] Wei, Colin, et al. "Theoretical Analysis of Self-Training with Deep Networks on Unlabeled Data." International Conference on Learning Representations. 2020.

**Summary Of The Paper:**

This paper provides an in depth empirical evaluation of classical self-training techniques such as pseudo-labelling and entropy minimization on test performance under domain shifts. The authors stress that, although simple, these techniques consistently improve the robustness to distribution shifts regardless of model architecture or pre-training techniques used. This makes them especially useful to practitioners applying machine learning algorithms to real problems where distribution shifts are prevalent. The authors claim state-of-the-art adaptation results on a number of popular dataset corruption benchmarks, and present a new challenging dataset for evaluating the robustness of deep vision models.




**Summary Of The Review:**

Overall I like the paper. I think it would bring value to the research community and it serves as a reminder that the simplest methods often work very well in practice.

---

> ### Author Response · Authors · 2021-11-23
> **Response to Reviewer s85k**
>
> Thanks a lot for your review and positive comments. We just posted a paper revision, especially adding additional references and discussions of related work along with new ablations/controls other reviewers asked for. Please let us know if you have additional questions or concerns that could improve the manuscript.
>
> > "The self-training algorithms evaluated here are well established, even in a deep learning setting"
>
> A major goal of this paper is to raise awareness for the effectiveness of simple self-learning methods, which we believe has huge value in practice, in particular in settings where UDA is not feasible.
> Self-learning as such has indeed been well known before; still, in the space of test-time adaptation methods, technically “new” techniques like Test-Time adaptation have been proposed, without testing conceptually more simple baseline techniques (such as variants of self-learning) before. Regarding these techniques, we show that a “novel” technique like TTT is actually easily outperformed by simple variants of self-learning (cf. Tables 4+22).
> We modified the introduction to better reflect our goal and added a Related Work section to better distinguish ourselves from other self-learning methods.
>
> > “Because there are so many results, the paper is quite dense and a bit hard to follow at times since the text is broken up a lot. This isn't a major issue but something the authors could consider improving for the final version.”
>
> We are fully aware of the challenge that comes with conveying such a large amount of material in a concise and clear manner. We have already put in a lot of effort into the presentation and into focusing only on the most relevant results in the main text, and we will continue to do so. Do you have suggestions for further improving the manuscript?
>
> > “Self-training using deep learning as presented here has gained popularity recently, and the paper is missing some related references (see below to name a few).”
>
> Thanks --- We updated the paper and included a Related Work section where we discuss the papers you and the other reviewers suggested. Due to space reasons, we had to put the more detailed version of the related work into the Appendix (section F).
>
> > “Reporting standard deviations of results would strengthen the author's claims.”
>
> To limit the compute, we ran RPL and ENT for our vanilla ResNet50 model three times with the optimal hyperparameters. The averaged results, displayed as “mean (unbiased std)” are:
>
> | ResNet50 + self-learning | mCE on IN-C dev [%] |mCE on IN-C test [%] |
> |--|--|--|
> | ENT | 50.0 (0.04) | 51.6 (0.04) |
> | RPL | 48.9 (0.02) | 50.5 (0.03) |
>
> We included these results in Appendix C.8, Table 24.
>
> While we would prefer running all experiments multiple times and reporting averaged results, this is unfortunately unfeasible given the high computational requirements for the IN scale experiments, especially on larger models---note that the results are generally already averaged across the 75 IN-C corruptions, and we provide full results on all model variants in the appendix for individual corruptions.

---

> > ### Comment · Reviewer_s85k · 2021-11-24
> > **Response**
> >
> > > A major goal of this paper is to raise awareness for the effectiveness of simple self-learning methods, which we believe has huge value in practice, in particular in settings where UDA is not feasible. Self-learning as such has indeed been well known before; still, in the space of test-time adaptation methods, technically “new” techniques like Test-Time adaptation have been proposed, without testing conceptually more simple baseline techniques (such as variants of self-learning) before. Regarding these techniques, we show that a “novel” technique like TTT is actually easily outperformed by simple variants of self-learning (cf. Tables 4+22). We modified the introduction to better reflect our goal and added a Related Work section to better distinguish ourselves from other self-learning methods.
> >
> > Yes, like I said in the main review I think this paper serves as a reminder that the simplest methods often work well in practice. With that said, from a technical standpoint I still stand by my initial statement in that the technical novelty is somewhat limited here. Nonetheless, I believe the overall message and empirical study presented in this paper is valuable for the community.
> >
> > > We are fully aware of the challenge that comes with conveying such a large amount of material in a concise and clear manner. We have already put in a lot of effort into the presentation and into focusing only on the most relevant results in the main text, and we will continue to do so. Do you have suggestions for further improving the manuscript?
> >
> > Sure, I would for example try to dedicate pages 5/6 to include only results tables, such that no text is broken-up in between.
> >
> > > While we would prefer running all experiments multiple times and reporting averaged results, this is unfortunately unfeasible given the high computational requirements for the IN scale experiments, especially on larger models---note that the results are generally already averaged across the 75 IN-C corruptions, and we provide full results on all model variants in the appendix for individual corruptions.
> >
> > Thanks for the additional results. I understand there are compute constraints at this stage, if we'd had access to some basic statistical significance it would have strengthened the empirical claims of the paper. Perhaps it's something to plan for ahead of time in the future.

---

### Official Review · Reviewer_PUq6 · 2021-11-02

**Correctness:** 3
**Technical Novelty And Significance:** 2
**Empirical Novelty And Significance:** 2
**Recommendation:** 5
**Confidence:** 4

**Main Review:**

Strengths
- The paper shows robust results where adding self-learning on top of existing methods (by starting from the checkpoint of the previous methods) improves performance, even when the existing method has already seen the unlabeled target data (UDA methods). One interesting test to add here would be where self-learning helps on top of using self-learning for UDA (like applying Noisy student on source and target unlabeled data). It seems to be complementary to doing self-learning (Noisy student) on the ImageNet tasks.
- There are interesting results on how self-learning improves large pre-trained models beyond BN adaptation. However, a comparison to other methods is missing here.
- As a pretty generic method, self-learning also seems to improve over test-time adaptation approaches like Test-time Training which had a more similar setting in mind. This shows that the gains are not purely from the setup of test-time adaptation.
- The paper is pretty comprehensive in methods and architectures, also testing on recent self-supervised methods (DINO).
- There are some interesting experiments on the proposed variant of DomainNet they call ImageNet-D, which tests transfer from ImageNet to overlapping classes in DomainNet. DomainNet allows for domain-level insights, breaking down where self-learning fails. However, I am left wondering whether ImageNet to ImageNet-D is different from the Real -> any other domain part of DomainNet. Perhaps the main gains here are that the scale of the Real data is much larger and that we can test models that are trained on ImageNet rather than a different special dataset.
- The paper gives an interesting preliminary analysis of the self-learning dynamics in a simple two-point setting, with some results for setting student and teacher temperatures in self-learning that seem to bear out in practice as well. The result seems general beyond test-time adaptation.

Weaknesses
- There have been quite a few works that focus on how self-training / self-learning improves distribution shift and how self-training and pre-training stack together: https://arxiv.org/abs/2012.04550, http://proceedings.mlr.press/v139/cai21b.html (in the experimental section), https://arxiv.org/abs/2002.11361, https://arxiv.org/abs/2106.04732, https://arxiv.org/abs/2006.10032, https://arxiv.org/abs/2012.11460, https://arxiv.org/abs/2006.06882 (this one is just about stacking them, not distribution shift). In some sense, the main premise of the paper (for example the title) is somewhat well-established. Discussion about these and your distinction from them would improve the positioning of the paper.
- The setting is a little confusing at first - it would seem that from an "access" point of view, we should always have access to the source dataset, while we may not be able to know the target dataset. I think this comes from the way that it's explained in the context of UDA. From what I understand, the main thing with the "source-free" setting is that the source dataset is too large so we need to do pre-training. I think its a bit more straightforward to start with the OOD generalization setting (what you call ad-hoc) without any knowledge of the target, then add the knowledge of the target data. Also, the "test-time adaptation" term has strong connotations of the online setting, which I believe is not being considered here, but should be distinguished clearly.
- Fig 1 clarity: Overall, the figure doesn't say very much about the source-free setting (it just looks like an option, but pictorially it doesn't look different from the other options). Some picture-level aspects were unclear, like what the different between the icons in the gray (pre-training) box are, and what the image + 3 gray shapes in the top represent, as well as what the orange stripe in the 3 gray shapes in the adaptation phase denote.
- Table 1,2 seem to use different self-learning algorithms. How were the methods for each table chosen? In Section 6, the paper supports Robust pseudo labeling, but Table 2 uses ENT for the main results.
- In Table 4, why is one comparison against TENT while the other against TTT?
- The takeaway in Table 8 that updating all affine layers is important is perhaps misleading, since in Table 5, tuning all affine layers is the worst option. It is also not fully specified what tuning all affine layers means?
- It may be easier in terms of presentation to talk about ImageNet-D right before the results in section 7.
- The results on ImageNet-D so far do not seem unique to test-time adaptation with self-learning; they would seem to occur in other settings too. Are there any aspects of ImageNet-D that test aspects of test-time adaptation methods?
- As the paper suggests ImageNet-D as a robustness benchmark, it's probably worth discussing whether we would expect our models to generalize to such disparate domains - for some of the domains, the paper reports close to 0% accuracy.

**Summary Of The Paper:**

The paper considers using self-learning / self-training as a test-time adaptation step for adapting to distribution shifts, showing that it is complementary with pre-training methods, domain adaptation methods used for training the model. The test-time adaptation setting (source-free domain adaptation setting) is taken to be the setting where models can use target data to adapt but cannot use the source data (which was used for pre-training).

**Summary Of The Review:**

The paper tests self-learning as a complementary addition to improve robustness. However, the premise that self-learning improves robustness is already somewhat well-established - the main contribution here is a systematic application to different methods and datasets. The restriction to the pre-training + test-time adaptation setting has also been considered to some extent, but not as systematically. The value of the proposed dataset ImageNet-D is unclear, whether it gives insights beyond DomainNet itself, and whether it is a worthwhile goal to generalize to such disparate domains. Finally, the analysis of self-learning dynamics seems interesting and predicts some empirical behaviors nicely. I think the paper could have a good message solidifying self-learning methods for robustness, but could use some tightening up in the story/clarity of the paper, and some inconsistencies in the experimental reporting. I'd be happy to raise my score if the issues are addressed in the rebuttal.

---

> ### Author Response · Authors · 2021-11-23
> **New Result: Can self-learning improve over self-learning based UDA?**
>
> Thanks a lot for your review and the idea of testing self-learning on top of self-learning based UDA  --- we also included the following result in section E.3 in the paper. We will keep this separate from our full response to be able to include the result table here.
>
> ### Does the proposed self-learning method help on top of self-learning based UDA methods? For example, one could continue with the Noisy student training on source and target unlabeled data.
>
> We agree that this is an interesting setting --- a limitation is the required compute budget. We still investigated this question in the revised paper, by considering the smaller scale UDA setup in French et al. (2018), which might be most similar to your proposed setting.
>
> We reproduced the small scale results from their paper and adapted all models using self-learning, with different data augmentation techniques on the student model. We find that this UDA setup is indeed already “optimal” as we already hypothesized in our introduction on UDA, and cannot be meaningfully improved with additional self-learning during test time. In the Table below, we report accuracy (higher is better).
>
> | | Baseline | BN adapt | ENT | RPL | Self-Ensembling Loss |
>  |--|--|--|--|--|--|
>  | mnist->svhn |
> MT+TF | 33.88 | 34.44 | 34.87 | 35.09 | 33.27 |
> MT+CT* | 32.62 | 34.11 | 34.25 | 34.21 | 33.36 |
> MT+CT+TF | 41.59 | 41.93 | 41.95 | 41.95 | 42.7  |
> MT+CT+TFA | 30.55 | 32.53 | 32.54 | 32.55 | 30.84 |
> SVHN specific aug.| 97.05 | 96.82 | 96.91 | 96.87 | 97.12 |
>  | | | | | | |
>  | mnist->usps |
> MT+TF | 98.01 | 97.91 | 97.96 | 97.91 | 98.16 |
> MT+CT* | 88.34 | 88.39 | 88.54 | 88.39 | 88.44 |
> MT+CT+TF | 98.36 | 98.41 | 98.41 | 98.41 | 98.5  |
> MT+CT+TFA | 98.45 | 98.45 | 98.45 | 98.45 | 98.61 |
>  | | | | | | |
>  | svhn->mnist |
> MT+TF | 98.49 | 98.47 | 98.49 | 98.47 | 99.4  |
> MT+CT* | 88.34 | 88.36 | 88.36 | 88.36 | 89.36 |
> MT+CT+TF | 99.51 | 99.49 | 99.5  | 99.49 | 99.57 |
> MT+CT+TFA | 99.56 | 99.57 | 99.57 | 99.57 | 99.58 |
> SVHN specific aug. | 99.52 | 99.49 | 99.5  | 99.49 | 99.65 |
>  | | | | | | |
>  | usps->mnist |
> MT+TF | 92.79 | 92.62 | 92.62 | 92.66 | 93.08 |
> MT+CT* | 99.11 | 99.13 | 99.14 | 99.13 | 99.21 |
> MT+CT+TF | 99.41 | 99.42 | 99.45 | 99.42 | 99.52 |
> MT+CT+TFA | 99.48 | 99.54 | 99.57 | 99.54 | 99.54 |
>
> This “full UDA” setup is however much more expensive on ImageNet scale and would require 90 epochs of training on 1.25M images (1.2M train, 50k dev) for each single corruption in ImageNet-C, in contrast to 90 epochs of training on clean ImageNet, saving the checkpoint, and running 5 epochs of adaptation on the 50k test images for each domain.
> Adapting with a noisy-student-like setup might however be interesting --- during early experimentation, we ran an experiment where we tested SimCLR [1] and AugMix [2] augmentations on the student network but could not find a meaningful improvement over our current adaptation scheme. This might be due to the choice of dataset (ImageNet-C).
>
> We updated the paper in this regard and added our new UDA results to Appendix E.3. Thanks for raising this interesting discussion point, and please let us know if our response and paper update address your concern.
>
> [1] Chen et al., “A Simple Framework for Contrastive Learning of Visual Representations”
>
> [2] Hendrycks et al., “AugMix: A Simple Data Processing Method to Improve Robustness and Uncertainty”

---

> ### Author Response · Authors · 2021-11-23
> **Response to Reviewer PUq6 (Part I)**
>
> Thank you for your thorough and insightful review. We will address your specific concerns below. We also updated the paper accordingly.
>
> > “There are interesting results on how self-learning improves large pre-trained models beyond BN adaptation. However, a comparison to other methods is missing here.”
>
> To the best of our knowledge, the most relevant techniques which are comprehensively evaluated on ImageNet-C and other ImageNet scale datasets are Test-Time Training, TENT, and batch norm adaptation. If you are aware of other relevant techniques we should compare to, we very much appreciate a hint.
>
> > There have been works discussing how self-training / self-learning improves distribution shift and how self-training and pre-training stack together which should be discussed.
>
> Thanks --- We updated the paper and included a Related Work section where we discuss the papers you and the other reviewers suggested. Due to space reasons, we had to put the more detailed version of the related work into the Appendix (section F).
>
> > “The setting seems a little confusing at first. From an "access" point of view, we should always have access to the source dataset, while we may not be able to know the target dataset. Are we interested in the "source-free" setting because the source dataset is too large so we need to do pre-training? One should rather motivate the approach with the OOD generalization setting without any knowledge of the target, then add the knowledge of the target data.”
>
> We agree and have updated the introduction accordingly.
> UDA-type methods have several issues in practice which we attempt to overcome: First, they require retraining the classifier from scratch if the domain changes and second, they require access to the source domain. That renders UDA-type methods often useless in practical scenarios. For example, an autonomous car cannot be retrained from scratch if the weather changes, even if it had access to the source dataset it has initially been trained on. An image-based quality control software may not necessarily open-source the images it has been trained on, but still has to be adapted to the lighting conditions at the operation location. Finally, the Noisy Student model, which has the best reported numbers on a variety of image benchmarks, has been trained on the Google internal JFT dataset, which neither we nor the general public have access to.
> Due to the issues associated with UDA, these types of applications have been addressed by ad-hoc robustness methods so far (= no adaptation during test time). We believe that the next step would be to utilize simple source-free, minimal compute requiring domain adaptation solutions. We have rephrased our introduction accordingly and put the focus more on the robustness application side instead of UDA.

---

> > ### Comment · Reviewer_PUq6 · 2021-11-27
> > **Response to authors**
> >
> > Thanks for the detailed response and clarifications. My main concern is this: self-learning as a method for distribution shift is well established and should probably be a baseline for robustness. I think it's nice that a simple self-training method works in this source-free regime, but overall, the premise of the paper isn't very surprising given the extensive work on self-learning / self-training for distribution shifts. In the related works section of the revised paper, the main difference between this paper and the other papers mentioned is in the "source-free" domain adaptation setting. However, the other methods that use unlabeled data (which generally include a labeled loss and unlabeled loss) can still be applied here by applying the labeled loss when we have the source data, then the unlabeled loss when we only have target unlabeled data. Thus I would consider works that use unlabeled data (VAT, Fixmatch/Adamatch, and other linked works from the original review) as methods to benchmark, beyond the "test-time adaptation" methods like TTT, TENT, etc.

---

> > > ### Author Response · Authors · 2021-11-27
> > > **Re: Response to authors**
> > >
> > > We fully agree that for someone coming from UDA and semi-supervised learning, self-learning for distribution shifts is well established and we are happy to cite associated UDA methods like AdaMatch much more prominently in the paper. The goal of our paper, however, is two-fold:
> > >
> > > First, we show that in the source-free setting, self-learning still works as fabulously as you’d expect. This might not be overly surprising, but to our knowledge isn’t well established, also not in the additional references we now included. We are not aware of works using the unlabeled loss part of Adamatch/Fixmatch/VAT in the source-free test-time adaptation setting, but testing them in addition to ENT or RPL would certainly be a great extension of our work and feed into our paper story. We will add results to the final version of the paper.
> > >
> > > Second, our paper aims to make the robustness community, which tends to focus on the source-free setting due to its higher relevance in practice, more aware that simple self-learning techniques which are at the core of many UDA methods, should be used as baselines instead of reinventing the wheel. Within the robustness (unlike the UDA) community, the effectiveness of self-learning is arguably less well established.

---

> ### Author Response · Authors · 2021-11-23
> **Response to Reviewer PUq6 (Part II)**
>
> > “Also, the "test-time adaptation" term has strong connotations of the online setting, which I believe is not being considered here, but should be distinguished clearly.”
>
> We agree and added this sentence to our introduction (see our revised paper): ”We here do not consider test-time adaptation in an online setting like is studied e.g., in [1], where the model is adapted to one example at a time, and reset after each example.”
> However, we note that our approach is compatible with a “continuous” domain adaptation setting where the model gets a continuous stream of data from the same domain and the model performance improves continuously with more data: https://i.postimg.cc/LsMhVNBs/learning-curves.png. Here, we display the error of the Noisy Student L2 model while it is being adapted to ImageNet-C and ImageNet-R. We included and discussed the Figure in the link in Appendix C.9, Figure 7.
> [1] Zhang et al., “MEMO: Test Time Robustness via Adaptation and Augmentation”
>
> > Figure 1 is not clear.
>
> We agree and updated the figure. Please let us know if the updated figure addressed your concerns.
>
> > “Table 1,2 seem to use different self-learning algorithms. How were the methods for each table chosen? In Section 6, the paper supports Robust pseudo labeling, but Table 2 uses ENT for the main results.”
>
> To avoid clutter, we decided to show the best overall results in Tables 1+2. Thus, the best results on IN-C have been achieved with RPL, while the best results on CIFAR10-C have been achieved with ENT. The full results on IN-C are shown in Appendix C.3 and the results on the small scale datasets are shown in Appendix C.4. (It is an interesting observation that for all small scale datasets, ENT outperforms RPL, while on the large scale data, RPL outperforms ENT.)
>
> > In Table 4, why is one comparison against TENT while the other against TTT?
>
> The reason for this dichotomy is that the reference algorithms were not fully scaled to all considered datasets and methods: On ImageNet, TENT only showed results for a ResNet50, while TTT only showed results for a ResNet18 and only at severity level 5. This observation is consistent with your comment on missing baselines for adapting large-scale models---there truly are none to compare against, at least on the popular robustness IN-scale benchmarks. A full comparison of e.g. ResNet50 and larger models with TTT is out of scope (if it should be fair, i.e., with hyperparameter tuning), and in our opinion does not seem worthwhile, given that the conceptually simpler self-learning algorithms outperform the algorithm on the considered dataset.
>
> > “The takeaway in Table 9 that updating all affine layers is important is perhaps misleading, since in Table 5, tuning all affine layers is the worst option. It is also not fully specified what tuning all affine layers means?”
>
> Thanks for catching this. First, we apologize for the confusion; with “affine parameters”, we mean affine batch normalization (BN) parameters (i.e., the rescaling parameters beta and gamma). We clarified this point in the manuscript. In CNNs, we use the existing BN layers, and in Vision Transformers, we insert new affine layers, following the method introduced in [1].
> Additionally, the mechanism, i.e., which layers have to be optimized, is in fact architecture dependent. Full model tuning failed for ResNet50 / Resnext101 models and was not possible due to hardware constraints for the EfficientNet model. For a (much smaller) Resnet18, initial results show that full model adaptation actually performs slightly better -- at a low learning rate of 0.000075 at batch size 64, a ResNet18 can be improved from 69.5 % dev mCE to 63.5 % dev mCE with affine and 62.1% dev mCE with full adaptation. Still, our results on the standard ResNet50, and larger models suggest that the by far most robust technique is to adapt the BN layers when using CNN models trained with batch normalization.
> For Vision Transformer models, adapting all weights worked better than the other mechanisms. We changed the title of Table 9 to reflect that the take-away only applies to CNN architectures and that in Vision Transformers, full layer adaptation works better than affine layer adaptation.
>
> [1] Houlsby et al. “Parameter-Efficient Transfer Learning for NLP”

---

> ### Author Response · Authors · 2021-11-23
> **Response to Reviewer PUq6 (Part III, ImageNet-D)**
>
> > It may be easier in terms of presentation to talk about ImageNet-D right before the results in section 7.
>
> We agree with this suggestion and have merged the introduction to ImageNet-D from section 3 to the results section 6 (see revised paper).
>
> > What is the benefit of ImageNet-D over DomainNet?
>
> We filtered and reformatted the DomainNet dataset: DomainNet classes without a clear correspondence to ImageNet were discarded, and overlapping classes were remapped to the closest class in ImageNet. This allows robustness researchers to easily benchmark on this dataset, without the need of re-training a model (as is common in UDA). We made this point more clear in the introduction of the dataset, see the second paragraph in section 6.
> Does this properly address your concern?
>
> > “The results on ImageNet-D so far do not seem unique to test-time adaptation with self-learning; they would seem to occur in other settings too. Are there any aspects of ImageNet-D that test aspects of test-time adaptation methods?”
>
> Our suggestion would be to use ImageNet-D as an additional robustness benchmark, both in an ad-hoc and in an adaptation setting. We believe it is an interesting robustness dataset since some ImageNet-D domains seem to be quite challenging even to the best EfficientNet-L2 model (including the “Real” domain).
>
> > “As the paper suggests ImageNet-D as a robustness benchmark, it's probably worth discussing whether we would expect our models to generalize to such disparate domains - for some of the domains, the paper reports close to 0% accuracy.”
>
> We think that model behavior can be interesting to analyze even at an accuracy of 0%. As humans, we easily generalize to the difficult domains “Infograph” and “Quickdraw”. Even small children can classify animals in a picture book correctly without much supervision from similar picture books. Thus, a complete break-down of a model with a “super-human” performance on certain domains highlights the brittleness of the features it uses.
>
> We do not find it surprising that a vanilla ResNet50 does not generalize to “Quickdraw” or “Infograph”; however, the Noisy EfficientNet has been trained on 300M images and is the current state of the art on many robustness tasks. We believe that demonstrating its limitations is valuable and interesting.
>
> Additionally, we find it interesting to analyze which kinds of errors a model makes from a standpoint of interpretability.  The errors of our vanilla ResNet50 on ImageNet-D are well-interpretable, e.g., it systematically predicts the class “comic book” for images from the “Clipart” or “Painting” domains, “website” for images from the “Infograph” domain, and “envelope” for images from the “Sketch” domain. Thus, the classifier predicts the domain rather than the class. We find this behavior interesting and consider it to be a confirmation of the texture-vs-shape hypothesis [1]: The classifier uses texture cues rather than shape cues to make its decision.
>
> [1] Geirhos et al., “Imagenet-trained cnns are biased towards texture; increasing shape bias improves accuracy and robustness.”, ICLR 2019

---

### Official Review · Reviewer_b5KJ · 2021-11-02

**Correctness:** 3
**Technical Novelty And Significance:** 2
**Empirical Novelty And Significance:** 3
**Recommendation:** 6
**Confidence:** 4

**Main Review:**

Strength:
- The paper is very well written and organized and easy to follow. They also explain the target and overarching goal very clearly.
- The empirical study is relevant and beneficial for the research community.
- The proposed suggestion is easy to use/implement, and there are interesting results supporting the main claim.


Main concerns and comments to improve the paper:
- Real world evaluation beyond natural images is missing which helps to solidify claims of the paper: The paper focuses on improvement on curated datasets such as IN-family. There are prior works that show similar results including [Barret’20, Chen’20, and etc]  and premises of self-training have been established previously.  However the paper does not consider real-world applications such as test time performance drop in medical data or satellite images and only limited to curated datasets which limits the future impact of the work.
- Definition of mCE (mean Corruption Error) and the calculation procedure is required.
- The method is of an incremental nature and not novel.

Suggestion to improve the paper:
- Consider adding/studying other domain dataset and tasks. There are multiple open source datasets available.   You can check the WILDS benchmark.
- Adding model calibration and statistical analysis of the results can boost the validity of the result section.
- Consider using Big Transfer (BiT) model performance as one family of Models for ImageNet-scale datatsets.


**Summary Of The Paper:**

The paper studies effectiveness of self-training to improve test time performance when the distribution of test data is not similar to the training data. The paper more specifically focuses on source-free domain adaptation settings where the source target data is not available. In this setup self-training has been tested as an additional step on top of  different robustness and adaptation approaches such as robust pretraining, unsupervised domain adaptation and self-supervised pretraining. The paper shows improvement on multiple ImageNet variants and CIFAR10-C, and also introduces ImageNet-D dataset as a new benchmark. ImageNet-D has been produced by matching label space of IN datasets with DomainNet data provided in Visual domain adaptation challenges. The main contribution of this paper is to perform a systematic and large study of self-training as a method to deal with distribution shifts.

**Summary Of The Review:**

The paper ran a large scale study to establish the benefits of self-training for data distribution shift. The technical contributions of the paper are only marginal and more of incremental nature, however the study itself is valuable and can be beneficial for the community. This study can significantly get boosted by diversifying the range of datasets and tasks under study.

---

> ### Author Response · Authors · 2021-11-23
> **Response to Reviewer b5KJ**
>
> Thanks a lot for your review. We uploaded a full paper revision to address your suggestions about new experiments and ablation studies. Please let us know whether your concerns are adequately addressed.
>
> ### Definition of mCE (mean Corruption Error) and the calculation procedure is required.
>
> Thanks for catching this --- we updated the paper and added the full definition to the Appendix. We added the following sentence to the main text (see updated paper version): “The established metric to report model performance on IN-C is the mean Corruption Error (mCE) where the error is normalized by the AlexNet error, and averaged over all corruptions and severity levels, see Eq.20, Appendix C.1.” We could also move equation 20 to the main paper if you prefer. Does this address your concerns?
>
> ### The method is of an incremental nature and not novel.
>
> The major goal of this paper is to raise awareness for the effectiveness of simple self-learning methods, which we believe has huge value in practice, in particular in settings where full UDA is not feasible.
> Self-learning as such has indeed been well known before; still, in the space of test-time adaptation methods, technically “new” techniques like Test-Time Training have been proposed, without testing conceptually more simple baselines (such as variants of self-learning) before. Regarding these techniques, we show that a “novel” technique like TTT is actually easily outperformed by simple variants of self-learning (cf. Tables 4+22).
> We modified the introduction to better reflect our goal and added a Related Work section to better distinguish ourselves from other self-learning methods.
> Also see our response on discussing the various pointers to related work from all reviewers --- which again highlights that while self-learning has been considered on many problems, application to test-time adaptation at scale has not yet been considered.
>
> ### Suggestions on additional experiments to improve the paper
>
> Thanks for the great suggestions for additional experiments and analyses. We performed experiments on WILDS and on BiT models and updated the paper (but had to put all result tables to the supplement for space reasons).
> Please let us know if you have additional comments about evaluations.
>
> #### WILDS benchmark
>
> On WILDS, self-learning is effective on the systematic distribution shift in the Camelyon17 benchmark, and less effective on the “mixed” distribution shifts on datasets like RxRx1, PovertyMap, and FMoW. We think that the result supports our claim that self-learning is effective, while showing the important limitation when applied to more diverse shifts. Thanks for the suggestion! In the Table below, we report accuracy (higher is better).
>
> | | Validation (ID) | Validation (OOD) | Test (OOD) |
> |-- |-- |-- | --|
> | Camelyon17| | | |
> | Baseline  |  81.4 |  88.7 |  63.1 |
> | BN adapt  |  97.8 (+16.4)  |  90.9 (+2.2)  |  88.0 (+24.9)  |
> | ENT  |  97.6 (+16.2)  |  92.7 (+4.0)  |  91.6 (+28.5)  |
> | GCE  |  97.6 (+16.2)  |  93.0 (+4.3)  |  91.0 (+27.9)  |
> | | | | |
> | RxRx1| | | |
> | Baseline  |  35.9 |  19.1 |  29.7 |
> | BN adapt  |  35.0 (-0.9)  |  19.1 (0.0)  |  29.4 (-0.3)  |
> | ENT  |  34.8 (-1.1)  |  19.2 (+0.1)  |  29.4 (-0.3)  |
> | GCE  |  34.8 (-1.1)  |  19.2 (+0.1)  |  29.4 (-0.3)  |
> | | | | |
> | FMoW| | | |
> | Baseline  |  60.5 |  59.2 |  52.9 |
> | BN adapt  |  59.9 (-0.6)  |  57.6 (-1.6)  |  51.8 (-1.1)  |
> | ENT  |  59.9 (-0.6)  |  58.5 (-0.7)  |  52.2 (-0.7)  |
> | GCE  |  59.8 (-0.7)  |  58.6 (-0.6)  |  52.1 (-0.8)  |
> | | | | |
>
>
> ### Big Transfer (BiT) models
>
> In the rebuttal time frame, we ran an initial study on the distilled BiT ResNet50 model in the TIMM repository and found a small improvement from 55.0% to 54.4% test mCE on ImageNet-C.
> The BiT models use group normalization instead of batch normalization. While affine adaptation works well when batch normalization is used, it is not clear whether this would be the optimal adaptation mechanism in models using group normalization. For example, we find that in Vision Transformer models, adapting all parameters works better than adapting the introduced affine layers, see Table 5 in our paper.
> We think that BiT models are an interesting addition to the paper and will run additional studies. Please see E.2 for the full results so far.
>
>
> ### “There are prior works that show similar results including [Barret’20, Chen’20, and etc] and premises of self-training have been established previously.”
>
> Thanks --- We updated the paper and included a Related Work section where we discuss the papers you and the other reviewers suggested. Due to space reasons, we had to put the more detailed version of the related work into the Appendix (section F).

---

> ### Author Response · Authors · 2021-12-06
> **Thanks for your review / summary comment**
>
> Dear reviewer **b5KJ**, thanks again for your comments. We [revised our manuscript](https://openreview.net/forum?id=1oEvY1a67c1&noteId=dmin7DraT-p) to address your main concerns. Especially your comment “This study can significantly get boosted by diversifying the range of datasets and tasks under study” encouraged us to run a range of additional control experiments, and test the method in new settings and tasks. These added results (on [WILDS, BiT](https://openreview.net/forum?id=1oEvY1a67c1&noteId=dmin7DraT-p), [Self-ensembling](https://openreview.net/forum?id=1oEvY1a67c1&noteId=9b3i7yuqdhF)) realistically point out additional strengths and limitations of self-learning.
>
> Please let us know if your concerns are adequately addressed ─ we are happy to additionally adapt the manuscript. We would of course greatly appreciate it if you would reconsider your score based on our rebuttal and added experiments.

---

### Author Response · Authors · 2021-11-23
**Overall response and summary of changes**

We thank all reviewers for their valuable feedback & appreciating easiness of implementation (**b5KJ**), interesting results (**b5KJ**, **PUq6**), thorough, significant and convincing empirical evaluation (**s85k**), their appreciation of the relevance of this paper to the robustness research community (**b5KJ**, **s85k**) and clear writing and organization (**b5KJ**, **PUq6**, **s85k**).

We just posted a revised version of the paper to address open questions and concerns raised by the reviewers.
The pdfdiff is accessible [here](https://openreview.net/revisions/compare?id=1oEvY1a67c1&left=3p79MyO-8gX&right=_13pnUb9lu&pdf=true).

### Changes to the intro and additional discussion of related work.

We added additional discussion of related work, and adapted our introduction based on the feedback of all reviewers. Supplementary section F now contains an in-depth discussion of additional references. We also updated Figure 1 to better outline the approach.

### WILDS benchmark

We included results on the WILDS benchmark in Appendix E. Self-learning is effective on the very systematic shift encountered on medical images of the Camelyon17 dataset, but less so on other datasets in WILDS (RxRx1, FMoW and PovertyMap) where less images per test domain are available, and multiple domains are shuffled.

We think that this result strengthens our motivation (with medical imaging being one prominent application example), while highlighting the importance of a systematic domain shift (as encountered in ImageNet-C, -R, etc.).

We thank **b5KJ** for the suggestion, and appreciate any additional feedback.

### BiT models

Following the suggestion of reviewer **b5KJ**, we tested self-learning on Big Transfer (BiT) models and included proof-of-concept results in Appendix E showing small gains on ImageNet-C with robust pseudo labeling.

### Self learning on top of UDA models based on self-learning

Reviewer **PUq6** pointed towards the application of self-learning to already domain-adapted models. We re-considered the setup in French et al (2017) and confirmed that self-learning at test time does not further improve UDA models already using self-learning at training time.

The results strengthen our point in the introduction (“This evaluation scenario [UDA] provides optimal conditions for adaptation” …) and it might be the case that also ImageNet-C evaluation would benefit from such a scenario --- however this training setup would require 75 full training runs on ImageNet scale just for a single evaluation, and is impractical.


### Additional updates

- Based on reviewer feedback, we updated Figure 1.
- We updated the plots for simulating the two point model, and added an ablation for the effect of different optimizer settings to the appendix, Fig 3--5. We also updated Fig 2 accordingly.

We discuss additional updates in the individual responses to all reviewers.

---

### Author Response · Authors · 2021-11-30
**Summary at the end of the discussion phase**

We thank all reviewers for their helpful comments. In summary, our changes are:

[**PUq6**] The introduction now focuses more on extending the ad-hoc robustness setting with self-learning and less on UDA. [In our ongoing discussion with PUq6, we also clarified this focus.](https://openreview.net/forum?id=1oEvY1a67c1&noteId=zK4XWp0hqEV)

[**PUq6**] Figure 1 has been modified to improve clarity.

[**b5KJ**, **PUq6**, **s85k** ] A short related work section has been added to Section 1 discussing the paper suggestions by the reviewers. A longer version has been added in Appendix F.

[**b5KJ**] A definition of the mCE has been added to Section 3.

[**PUq6**] An explanation of “affine” layers has been added to Section 5. Differences in adaptation mechanisms between CNNs and Vision Transformers are now discussed.

[**s85k**] We included averaged results for the mean and standard deviations across three seeds for a ResNet50 in Appendix C.8.

[**PUq6**] New Figure 7 shows self-learning for a continuous adaptation on ImageNet-C and ImageNet-R over a single epoch.

[**b5KJ**] Results for self-learning on WILDS have been added in Appendix E.1. The results suggest applicability to e.g. systematic domain shifts in medical imaging (Camelyon17), but not for more mixed domain shifts, which fits the considered setup we tested with ImageNet-C.

[**b5KJ**] Results for adapting Big Transfer models have been added in Appendix E.2.

[**PUq6**] Results for adapting models trained [with self-learning based UDA](https://openreview.net/forum?id=1oEvY1a67c1&noteId=9b3i7yuqdhF) have been added as an additional control in Appendix E.3.

[**Puq6**] We moved the introduction to ImageNet-D to section 6 where the results on ImageNet-D are shown to improve clarity and the paper flow. We also added a paragraph discussing the benefits of ImageNet-D over DomainNet in the context of robustness evaluation.

---

### Decision · Program_Chairs · 2022-01-20

**Decision:**

Reject

**Comment:**

The main contribution of the paper is to perform a systematic and large study of self-training as a method to deal with distribution shifts. Reviewers have appreciated the clarity in the overall writing of the paper, and rigor in the empirical analysis. However the main concern from two of the reviewers is that the technical contributions of the paper are only marginal and incremental in nature. The premise that self-learning improves robustness is already somewhat well-established (Reviewer PUq6 has pointed out papers that focus on how self-training / self-learning improves distribution shift and how self-training and pre-training stack together), and the main contribution of the paper is a systematic application to different datasets. Given the existing work on the relevance of self-training in distribution shift, the paper falls below the acceptance bar for ICLR in my opinion.